

# Air traffic and contrail changes during COVID-19 over Europe: A model study

Ulrich Schumann[1], Ian Poll[2], Roger Teoh[3], Rainer Koelle[4], Enrico Spinielli[4], Jarlath Molloy[5], George S. Koudis[5], Robert Baumann[1], Luca Bugliaro[1], Marc Stettler[3], Christiane Voigt[1,6]

[1]Institute of Atmospheric Physics, Deutsches Zentrum für Luft- und Raumfahrt, 82234 Oberpfaffenhofen, Germany
[2]Emeritus Professor of Aerospace Engineering, Cranfield University, UK
[3]Centre for Transport Studies, Department of Civil and Environmental Engineering, Imperial College, London, SW7 2AZ, UK
[4]Performance Review Unit, EUROCONTROL, 1130 Brussels, Belgium
[5]NATS, Whiteley, Fareham, Hampshire, PO15 7FL, UK.
[6]Johannes Gutenberg-University, Mainz, Germany

*Correspondence to*: Ulrich Schumann (Ulrich.schumann@dlr.de)

**Abstract.** The strong reduction of air traffic during the COVID-19 pandemic provides a test case for the relation between air traffic density, contrails, and their radiative forcing of climate change. Air traffic and contrail cirrus changes are quantified for a European domain for March to August 2020 and compared to the same period in 2019. Traffic data show a 72 % reduction in flight distance compared with 2019. This paper investigates the induced contrail changes in a model study. The contrail model results depend on various 20  methodological details tested in parameter studies. In the reference case, the reduced traffic caused an even stronger reduction in contrail length, partly because the weather conditions in 2020 were less favourable for contrail formation than in 2019. Contrail coverage over Europe with an optical depth larger than 0.1 decreased from 4.6 % in 2019 to 1.4 % in 2020; total cirrus cover amount changed from 28 to 25 %. The reduced contrail coverage caused 70 % less longwave and 73 % less shortwave radiative forcing 25  with the consequential reduction of 54 % in the net forcing. The methods include recently developed models for performance parameters and soot emissions. The overall propulsion efficiency of the aircraft is about 20 % smaller than estimated in earlier studies, resulting in 3 % fewer contrails. Considerable sensitivity to soot emissions is found highlighting fuel and engine importance. The contrail model includes a new approximate method to account for water vapor exchange between contrails and 30  background air and for radiative forcing changes due to contrail-contrail overlap. The water vapor exchange reduces available ice supersaturation in the atmosphere, which is critical for contrail formation. Contrail-contrail overlap changes the computed radiative forcing considerably. Comparisons to satellite observations are to be described in a follow-on paper.



# 1 Introduction

Contrails induced by water vapor and soot emissions from aircraft flying in cold and humid air masses are responsible for a large part of the climate impact of aviation (Lee et al., 2021). Contrails cause positive longwave (LW) and negative shortwave (SW) instantaneous radiative forcing (RF) components at top-of the atmosphere (TOA) (Meerkötter et al., 1999). A positive net effect (sum of LW and SW contributions) induces a warming of the Earth-atmosphere system. Contrails and aircraft engine emissions interact with the atmosphere in a complex

manner and not all aspects are well understood (Voigt et al., 2017; Kärcher, 2018; Lee et al., 2021). For example, contrails and aviation induced aerosols affect ambient cirrus clouds, potentially causing negative RF (Lee et al., 2010; Penner et al., 2018). Even for positive instantaneous RF, the effective radiative forcing controlling the global mean surface temperature is likely to be smaller than the instantaneous changes at TOA (Ponater et al., 2005; Rap et al., 2010; Bickel et al., 2020). One cannot exclude the possibility that contrail shadows cool the Earth's surface

regionally during daytime, while the all-day contrail greenhouse effect impacts the Earth surface more slowly, partly after weeks and longer, over larger domains and with possibly lower warming efficacy (Schumann and Mayer, 2017).

Contrails have been observed in many studies, but observational evidence for contrail warming is missing. This is because the expected changes are small, not well correlated with contrail cover and the observed changes may have

many causes (Minnis et al., 2004; Minnis, 2005; Schumann and Heymsfield, 2017). Only a few studies have related observed regional cirrus cover and TOA irradiance changes to contrails (Duda et al., 2004; Mannstein and Schumann, 2005; Stordal et al., 2005; Stubenrauch and Schumann, 2005; Haywood et al., 2009; Graf et al., 2012; Schumann and Graf, 2013; Spangenberg et al., 2013). Early attempts to relate contrails to reduced diurnal temperature changes associated with the air traffic changes over the USA in September 2001 (Travis et al., 2002)

were shown to be inconclusive, mainly because of the shortness of the period with reduced traffic (Dietmüller et al., 2008; Hong et al., 2008).

As a result of the COVID-19 pandemic, since March 2020 air traffic has experienced a strong, global, and long-lasting reduction (ICAO, 2021). In April 2020, worldwide air traffic reduced by 80 % compared to April 2019 according to aircraft transponder data collected by Flightradar24 (https://www.flightradar24.com/blog/march-

commercial-traffic-down-10-below-2019-so-far/). The European Organisation for the Safety of Air Navigation (EUROCONTROL) reported an almost 90 % decrease in air traffic over Europe for the same period, exhibiting a minimum in mid-April with a slow recovery thereafter (https://www.eurocontrol.int/covid19). The decrease was



significantly larger than the 17 % decrease of $CO_2$ emissions from global energy production in April 2020 compared to 2019 levels (Le Quéré et al., 2020).

This paper quantifies air traffic activity, the related contrail cirrus and the radiative forcing for Europe in the months March to August 2019 and 2020. A subsequent paper will compare the model results to satellite observations. The investigation within 20°W to 20°E and 35°N to 60°N covers much of Europe and the Eastern parts of the North Atlantic that have high air traffic density and are visible from the geostationary satellite METEOSAT (Schmetz et al., 2002).

The contrails are simulated with the Contrail Cirrus Prediction Model (CoCiP) (Schumann, 2012) which has been used for various related studies (Schumann et al., 2017; Voigt et al., 2017; Teoh et al., 2020b; Teoh et al., 2020a). The contrail model uses traffic data from EUROCONTROL for a large part of Europe and from the UK air navigation service provider NATS for the Shanwick Oceanic Control Area. Performance parameters, including fuel consumption and overall propulsion efficiency, are estimated using the Base of Aircraft Data (BADA 3) from
EUROCONTROL (Nuic et al., 2010; EUROCONTROL, 2015) and a recently developed open-access performance model for a set of subsonic turbofan-powered transport aircraft at cruise (Poll, 2018; Poll and Schumann, 2020a, b). Soot number emissions are computed with the fractal aggregate model (Teoh et al., 2019; Teoh et al., 2020b). The model uses numerical weather prediction forecast (FC) data from the European Centre for Medium Range Weather Forecasts (ECMWF) (Bauer et al., 2015). This paper describes the new traffic data set and its setup for
2019 and 2020. The data are used to quantify the changes in traffic, fuel consumption, soot emissions, contrail cover, RF and the related TOA irradiances. CoCiP is run in various model versions, to test the sensitivity of the results to model parameters, mainly in a new version approximating humidity exchange with background air and radiative contrail-contrail overlap, inspired by some earlier studies (Burkhardt and Kärcher, 2011; Schumann et al., 2015; Sanz-Morère et al., 2020). A set of parameter studies is presented that identify the model sensitivity to input
and model parameters. A further study of climatological effects and comparisons with satellite observation data, using simulations over a longer period is planned.

## 2 Air traffic and aircraft emissions input

This section briefly describes the data base of air traffic and aircraft emissions over Europe in 2019 and 2020 used for the contrail simulations. As a minimum, the input data must characterise the flights tracks and emissions in the
"upper" airspace above Flight Level (FL) 180 (18000 feet, about 5.5 km), where most contrails form. Previous CoCiP studies have used air traffic from various sources, including a global track data base for 2006 (Wilkerson et



al., 2010; Brasseur et al., 2016), data collected for the field experiment ML-CIRRUS over Europe and the North Atlantic in March/April 2014 (Schumann et al., 2016; Voigt et al., 2017) or traffic data for six weeks distributed over one year in 2012/13 in Japanese airspace (Teoh et al., 2020b). Here, all flights passing the European

investigation domain are considered. This includes all kerosene burning turbofan and turboprob engine aircraft. Piston engine powered aircraft only make a very small contribution to contrail formation. Input includes the aircraft type code, as defined by the International Civil Aviation Organization (ICAO), and the sequence of waypoints along the flight track. At each waypoint, the time, latitude, longitude, and FL, plus true air speed, instantaneous aircraft mass, fuel flow rate, overall propulsion efficiency and soot number emission index are specified, together

with a unique running flight number, and information on the traffic and the performance data sources used. The simulation code requires input in hourly sections with constant time resolution. The construction of this input starts with the list of flights inside or passing Europe, followed by the whole route from departure to the destination airport, which is required in order to estimate the aircraft take-off mass. This is then combined with meteorological wind and temperature data, and with performance and emission analyses, which is obtained from various sources

in a sequence of processing steps, see Supplement.

The prime sources for the aircraft position information are the so-called Correlated Position Report (CPR) messages provided by EUROCONTROL's Performance Review Unit (PRU). These data originate from the pan-European air traffic management system operated by EUROCONTROL (Niarchakou and Cech, 2019). The CPR represents augmented       surveillance       position       information,       based       on       real-time       surveillance       data

(https://www.eurocontrol.int/service/data-collection-service) derived from radar and from Automatic Dependent Surveillance - Broadcast (ADS-B) data (https://ads-b-europe.eu/). For flights outside the surveillance domain of EUROCONTROL, data from EUROCONTROL's so-called Model 3 (M3) data (Wandelt and Sun, 2015) are used, which contain partial track information from departure to destination also outside Europe. The M3 data are flight plan data partly corrected by surveillance (radar) data and are available from the DDR2 data repository of

EUROCONTROL. The M3 files provided by the PRU come directly from the network manager archives. For flights in the Shanwick control zone of the North Atlantic flight corridor, track information was provided by NATS. These were used to either replace or augment M3 data in that zone. The CPR data come without the ICAO aircraft type codes, but about 70 % of the CPR data contain the so-called ICAO 24-bit code, which is a unique aircraft identifier. A table relating many of the 24-bit codes with aircraft types was made available to us by Martin Schäfer

within OpenSky (Schäfer et al., 2014). In other cases, the type code was identified from the M3 or the NATS data for flights with same aircraft callsign, departure and destination locations and the same departure time.



For comparisons (see Supplement), aircraft position data as collected from a distributed net of ground-received ADS-B data and as purchased from Fligthradar24 AB, Sweden (https://www.flightradar24.com/how-it-works) were used. In addition, checks were performed by comparisons of the trajectory position data to position data obtained during flights of the DLR research aircraft. These confirmed that the position accuracy was in the order of 100 m in most cases.

Temperature and wind along the flight tracks are interpolated from global ERA5 reanalysis data, produced by ECMWF within the Copernicus Climate Change Service (Hersbach et al., 2020). Here, global 3-houly data with 1-degree geographic resolution, at pressure levels are used. True air speed is computed by subtracting the windspeed vector from the groundspeed vector. Temperature is required for computing Mach and Reynolds numbers and related aircraft performance parameters.

The contrail analysis requires information about the local aircraft mass, local fuel flow rate (in kg s⁻¹) and overall propulsion efficiency, together with water vapor mass and soot number emission indices per mass of burned fuel. Sensitivity studies with CoCiP show that a 10 % change in fuel consumption causes a change of about 7 % in contrail radiative forcing. Contrail formation depends on the overall propulsion efficiency, η, and an increase in η of 0.1 increases the threshold temperature by about 1.5 K (Schumann, 2000). Hence, more contrails form for larger η. Since most aircraft travel at temperatures about 5 to 12 K below the threshold temperature (Schumann and Heymsfield, 2017), the value of η has a smaller impact on the total mean contrail properties.

The fuel consumption rates and the overall propulsion efficiency are obtained from an aircraft performance model. In the past, the EUROCONTROL BADA3 model (Nuic et al., 2010; EUROCONTROL, 2015) was used for CoCiP studies (Schumann et al., 2011a). Alternatively, and in view of known limitations of the BADA3 method (Nuic et al., 2010), we use the self-contained and open source model "PS" presented recently (Poll and Schumann, 2020a, b), with a slight modification to allow for the full range of Mach numbers at cruise. The PS method has a more rigorous aerodynamic foundation and covers Reynolds number effects.

Fuel consumption rate is directly proportional to the aircraft mass, which is derived by subtracting the integrated fuel mass burned from the take-off mass. The take-off mass is the sum of the operational empty aircraft mass, the payload mass, and the total fuel mass. Unfortunately, take-off mass of aircraft is not recorded in publicly available data set. Consequently, take-off mass is estimated using an assumed payload load factor, LF (ratio of actual payload mass to maximum permitted payload mass). Data from the US Bureau of Transportation Statistics, from the German Statistical Federal Office, from EUROCONTROL and from ICAO (see Supplement) suggest lower passenger and



freight loadings after March 2020 than in the previous year (and more cargo flights). Therefore, LF is taken to be 0.7 for the time before the pandemic and 0.5 thereafter. The value 0.7 is found to be consistent with the actually flown FL profile staying below BADA3' estimate of the maximum altitude for the given mass (Eq. 3.5-1 (EUROCONTROL, 2015)) for most flights. The fuel mass is estimated from the total flight distance in air and

mean cruise aircraft performance. The overall propulsion efficiency, $\eta$, is defined as the product of engine net thrust and true air speed divided by the product of fuel flow rate and the lower calorific value of fuel (Cumpsty and Heyes, 2015). Both the fuel flow rate and the net thrust are provided by the performance model. The water vapor mass emission index and the lower calorific value of kerosene are set to 1.23 kg/kg and 43 MJ/kg, respectively.

Contrail properties are sensitive to the number of soot (or black carbon) particles emitted (Schumann et al., 2013a;

Kärcher, 2016; Burkhardt et al., 2018; Teoh et al., 2020b). For example, optical depth increases with the third root of the soot number emission index (Schumann et al., 2013a). The soot number emission index depends strongly on the engine type and operation state. Here, the emission index is computed for known engine types using engine data from the ICAO emission data bank and recently developed methods (Teoh et al., 2019). In the few cases when these data are not available, a constant soot number emission index of $10^{15}$ kg$^{-1}$ is assumed. The mean emission

index from this method is about $3 \times 10^{15}$ kg$^{-1}$, with large variability (Teoh et al., 2020b). With this emission index, the number of ice crystals per fuel mass burned in young contrails would be about a factor of two larger than observed (Schumann et al., 2013a). This may indicate a size or temperature dependent efficiency of soot particles acting as ice nucleus (Kärcher, 2016; Kleine et al., 2018; Lewellen, 2020). Therefore, the computed soot emission index value is halved in this study.

All these data are configured flight by flight, from departure to destination, without temporal interpolation and, finally, the flight tracks above FL 180 are split hourly and interpolated uniformly with 60 s time resolution. The resulting CoCiP input files require 36.8 GB (Gigabytes) of disk storage for March-August 2019 and 10.6 GB for the same period in 2020.

The mean traffic flight distances with respect to air (from true air speed and time, not over ground) and mean fuel

flow rates for the fleet of aircraft within the European investigation domain are listed in Table 1 for 2020 together with the percentage change relative to 2019. Figure 1 shows an example of the traffic tracks obtained from the various sources within two half-hour periods of 1 March 2020 (still "normal" traffic), one in the early morning with strong traffic from North America over the North Atlantic and one later in the morning with high traffic density over Europe. It can be seen that the CPR tracks are in good agreement with those from Flightradar24 (FR24).

Apparently, many aircraft were equipped with ADS-B receivers from which the FR24 data are derived. The NATS



data extend the CPR tracks in the Shanwick zone over the North Atlantic and the M3 data extend traffic in regions where surveillance data are missing.

As illustrated in Figure 2a, mean air traffic in upper airspace (above FL 180) over Europe decreased considerably after mid-March 2020. The total flight distance per day decreased by 72 % on average over the six-month period and by 91 % for the month of April in 2020 relative to 2019.

Figure 3 illustrates the spatial distribution of the mean traffic in terms of fuel consumption in the simulation domain for the six months on average in 2019 and 2020. Traffic and fuel consumption is largest along the route from London, UK, to Frankfurt/Main, Germany, but spreads along many other routes from the North Atlantic to the Near East and from Scandinavia to the Iberian Peninsula. Figure 3 also illustrates the large-scale traffic reduction in 2020 compared to 2019. The decrease of fuel consumption and flight distances are similar because the relative increase in aircraft weight (more cargo aircraft) is largely balanced by the lower load factor.

## 3. Numerical weather prediction data

Although 3-hourly ERA5 reanalysis pressure level data are used to provide the global traffic data with wind and temperature information, higher resolution deterministic operational numerical weather predictions from the Integrated Forecasting System (IFS) of the ECMWF (Bauer et al., 2015) are used for contrail simulation in the investigation domain. The IFS data are available for registered users. The IFS model used operates with a nominal resolution of 9 km horizontally, with 137 levels from the surface to model top at 0.01 hPa. Data are applied with 1 h time resolution and 0.25° horizontal geographic grid resolution. The mean vertical grid intervals in the IFS data between 200 and 300 hPa are about 10 hPa or 300 m for standard sea surface pressure. For comparison, the ERA5 data used are provided at fixed pressure levels, including 300, 250, 225 and 200 hPa, with vertical height intervals varying between 670 and 1200 m, i.e., with a much coarser vertical resolution. The forecast (FC) provide hourly three-dimensional fields of pressure, temperature, wind components, humidity, ice water content and cloud cover, plus two-dimensional fields for TOA irradiances of incoming solar direct radiation (SDR), reflected solar (RSR) and outgoing longwave radiation (OLR) on average over the recent hour.

A critical issue in the simulation of persistent contrails is the relative humidity (RHi) with respect to saturation over ice (Schumann, 1996; Irvine and Shine, 2015; Schumann and Heymsfield, 2017; Gierens et al., 2020). Here, RHi is derived from the FC data for temperature, pressure and absolute humidity with given water vapor saturation



pressure over ice (Sonntag, 1994). Several previous studies have found that ECMWF forecasts tend to underestimate the degree of ice supersaturation (Schumann and Graf, 2013; Kaufmann et al., 2018).

**Figure 4** compares the probability density function of relative humidity derived from the FC with data from ERA5 and the airborne in situ measurements on routine Airbus flights during the MOZAIC project (Petzold et al., 2020). Here, the FC and ERA5 data represent the RHi from interpolated temperature and absolute humidity along the flight tracks above Europe between 180 hPa and 310 hPa (about 12 and 8 km in the ICAO standard atmosphere) for the given time periods over Europe, while the MOZAIC data are from a longer time period and larger domain

at cruise levels of the Airbus A340, or A330 aircraft. Both NWP data sets underestimate the occurrence of high ice supersaturation. Part of this probably comes from the higher resolution of the measurements in time and space compared to the grid cell and hourly mean values provided by the numerical weather predictions. To avoid an underestimate of simulated contrails, in the past, CoCiP simulations usually were performed with enhanced humidity by dividing by a fixed model parameter $RHi_c \leq 1$. Previously, in order to obtain reasonable agreement

between model estimates and the observations (Schumann and Graf, 2013) large changes have been required (up to $1/RHi_c = 1/0.8 = 1.25$). However, more recently the forecast resolution has improved and so an $RHi_c$ equal to 0.95 is used in the reference cases and 1.0 and 0.9 in parameter studies. The potential contrail cover, i.e., the area fraction of air with temperature below the contrail threshold value and RHi > 100 % derived from the FC data amounts to 15 % at FL 350 (10.6 km) on average over the investigation domain for $RHi_c = 0.95$, which agrees with

estimates in the literature (Gierens et al., 2012) and shows that the selected $RHi_c$ value is reasonable.

While the results given in Figure 4 suggest that the quality of the ERA5 and FC data is about the same, the ERA5 data tend to underestimate wind shear, mainly because of the lower spatial resolution, see Figure 5. Wind shear is important for simulating contrail dispersion. Without dispersion, contrails would remain narrow, triggering ice clouds in the aircraft wake only (Lewellen, 2014; Paoli and Shariff, 2016). However, with shear and turbulence

driven dispersion, contrails grow in cross-section area and more and more contrail ice particles mix with ambient air, converting ambient ice supersaturation into contrail ice particles.

Another important parameter is the vertical wind. Adiabatic upward motion conserves mass specific humidity, but cools the air and, hence, enhances relative humidity, whilst downward motion reduces relative humidity (Gierens et al., 2012). The thickness of ice supersaturated layers, with relative humidity between ice saturation and liquid

saturation in raising air masses, increases for decreasing ambient temperature (Gierens et al., 2012). Therefore, vertical wind is controlling the persistence and lifetime of ice supersaturated air masses and contrails. Inspection



of several examples have shown that the ERA5 vertical wind is smoother in space and often smaller in magnitude than in the FC. Consequently, the FC data are preferred for contrail simulations.

Figure 6 gives an indication of the vertical depth of those layers suited to the formation of persistent contrails - as
derived from the FC data. The air temperature inside these layers is below the Schmidt-Appleman threshold value for contrail formation (for η = 0.35) and humid enough for persistency (RHi>1) (Schumann, 1996). The computed layer depth is limited by grid resolution and typically varies between 300 and 800 m, which is in the range of observations (Gierens et al., 2012). The values are largest over mountains because of frequent upgliding motions. Interestingly the thickness is larger over the North Atlantic than over the southern part of the domain. The geometric
thickness of layers with relative humidity between ice saturation and liquid saturation in raising air masses increases for decreasing ambient temperature (Gierens et al., 2012) and the air temperature is lower at higher latitudes. Hence the thicker layers over the North Atlantic may be partly because of lower air temperature. The thickness of the ice supersaturated layer limits the altitude range in which sedimenting ice particles persist and hence the thickness influences maximum ice water content reached in contrails (Lewellen, 2014; Schumann et al., 2015). This ice water
content and the geometrical depth also determine the optical thickness and, hence, the radiative forcing from contrails. Finally, the ice supersaturated layer thickness is important when discussing flight level changes to avoid warming contrails (Mannstein et al., 2005; Schumann et al., 2011a; Teoh et al., 2020a). Figure 6 also shows that the mean layer thickness over most of Europe was significantly larger in 2019 than in 2020, indicating that more contrails formed in 2019, not only because of more traffic, but also because of more favourable contrail formation
conditions.

## 4. Simulated contrail cover and related radiative forcing

The traffic, the emission input and the FC data described above are used for the contrail model CoCiP (Schumann, 2012). CoCiP simulates Lagrangian contrail segments from the initial formation in air satisfying the Schmidt-Appleman criterion (Schumann, 1996) until the final decay for each 60-s flight segment. The contrail physics
represented in this model is partly simplified compared to other models (Lewellen, 2014; Paoli and Shariff, 2016; Unterstrasser, 2016), but it resolves individual contrails and is applicable to global studies (Schumann et al., 2015). The model computes the local, contrail induced RF of each contrail segment for given contrail properties and given TOA solar and thermal irradiances using an algebraic model (Schumann et al., 2012) for an ice particle habit mixture (see Table 2 in Schumann et al. (2011b)) fitted to a set of reference data from libRadtran (Mayer and
Kylling, 2005; Emde et al., 2016). The code reads the meteorological data hourly, so that only two time slices are





kept in the core storage at a given time. Contrails surviving the hour are kept in a separate buffer in core memory and integrated in time over the next hour. The spatial distributions of contrail properties are evaluated each hour on a grid with about a 4.2 km mean horizontal resolution prepared for comparisons with Meteosat-SEVIRI observations (Schmetz et al., 2002) by summing the contributions from all the contrail segments, according to their

Gaussian plume properties. This gridded analysis consumes about 90 % of the computing time. Without this evaluation part and after the preparation of all the input data, the Fortran code takes less than 5 min on a laptop computer to run with traffic for the month of July 2019. The model parameters are set as described previously (Schumann et al., 2015), but including variable soot number emission index $EI_s$, humidity enhanced by a factor 1/RHic ( with RHic=0.95), plume mixing enhanced by differential radiative heating, contrail segments integrated

in the model's Runge-Kutta scheme with 1800 s time steps, and 10 h maximum contrail life time.

In regions of high traffic density, the amount of water entering contrails from ambient air may significantly dehydrate ambient air (Burkhardt and Kärcher, 2011; Schumann et al., 2015). Contrails take up water vapor from the ambient air and the first contrail formed reduces the ice supersaturation available for subsequent contrails flying later along about the same track (Unterstrasser, 2020). As explained in Sanz-Morère et al. (2020), contrail-contrail

overlap also affects the radiative forcing. When one contrail is formed, it changes the irradiances OLR and RSR at TOA. The RF is a function of these irradiances and reduced OLR and increased RSR values result in a smaller RF from the next contrail. A complete modelling of the humidity exchange and overlap effects would require integration of the prognostic equations for weather prediction and the related radiation transfer in time and space with resolution corresponding to the contrail scales. This is beyond the state of the art. Here, we account for

humidity exchange with background air and contrail-contrail overlap in an approximate manner. For each contrail, the mass of water vapor that enters as contrail ice is subtracted from the background field, and the mass of ice from the sublimating contrails is returned to the background humidity, conserving total water mass in the corresponding grid cell volume. To account for contrail-overlap in the RF analysis, the energy flux per grid cell area caused by the LW RF from a contrail is subtracted from the TOA OLR so that the RF from a subsequent overlapping contrail

is driven by a reduced TOA flux. This ensures that the effective OLR (after subtraction of LW RF) stays positive. For the SW flux, the albedo a=RSR/SDR is increased as a function of the SW RF, by |RF SW|/SDR. Here, SDR is the (incoming) solar direct radiation. This ensures that the increased albedo stays below one. These corrections are applied contrail by contrail in the sequence in which they occur in the traffic input and the changes in the background air and TOA irradiances are lost when reading the next FC input hourly. The effects are demonstrated

in the next section.



The contrail model has been applied and tested in several previous studies (Voigt et al., 2010; Schumann et al., 2011a; Jeßberger et al., 2013; Schumann and Graf, 2013; Schumann et al., 2013b; Schumann et al., 2013a; Schumann et al., 2015; Schumann et al., 2017; Voigt et al., 2017; Teoh et al., 2020b). Figure 7 demonstrates that the results from the improved method are both within the range of the previous results and within the scatter of
observation data for individual contrails. Without humidity exchange, the amounts of contrail ice, its particle sizes, optical depth and geometrical width and depth are between 10 to 30 % larger. These changes are within the range of scatter of the observations.

Figure 2 b-d show day-mean contrail properties and RF for the European domain as a function of time for the 6-month period. The contrail contributions vary strongly from day to day because of variable weather. The ratio of
contrail distance to flight distance is similar in both years, with a slight tendency to smaller ratios in 2020 because of the drier air. Similarly, the LW and SW RF values vary strongly and partially in anti-correlation. Hence, the day mean net RF is smaller, although positive on average. Some days with negative European mean net contrail RF are also found.

Figure 8 gives the mean optical depth of the sum of all contrails from the simulations for six months in 2019 and
the difference 2019-2020 and Figure 9 shows the net RF. Both are computed taking humidity exchange with background air and contrails overlap into account. The optical depth is seen to reach values up to 0.07 on average over these six months, with maximum changes of 0.054 between 2019 and 2020. However, it should be noted that this average includes contrail free days. Far larger values are reached in individual contrail segments – see Figure 7. The mean area-coverage of contrails with an optical depth larger than 0.1 decreased from 4.6 % in 2019 to 1.4
% in 2020. The mean cirrus cover in the domain in these periods reaches up to 28 % (see Table 1). Hence, the computed relative changes in cirrus cover are of the order of 10 % of mean cirrus cover.

The mean net RF varies from -0.2 to 0.8 W m$^{-2}$ over Europe and is mostly positive. Mean negative values occur over sea surfaces, mainly because of lower surface albedo than over land. Net RF values in 2020 are about 40 % lower than those in 2019. Hence, the reduction in net RF (60 %) is smaller than the reduction in traffic (72 %). This
is due, in part, to different changes of SW and LW RF and to the nonlinear effects from contrail-background humidity exchange and contrail-contrail overlap.

Finally, data are shown that may be compared with satellite observations in a follow-on study. These are optical depth (OT), OLR and RSR from the sum of cirrus and contrails. The OT presented in Figure 8 is sum of the OT of cirrus from the FC data and the OT from contrails computed with CoCiP. Here, the OT of cirrus without contrails





is estimated from the weather model output as a function of ice water content and temperature with effective ice
particle diameters parameterized from observations at -81°C to 0°C temperatures (Heymsfield et al., 2014). The
OLR given in Figure 11 is from the FC data minus the LW RF from contrails and the RSR in Figure 12 is from the
FC data minus the SW RF from contrails. We see large spatial variability of cirrus OT and the irradiances. The
variability is largest for RSR because of changes in cloudiness, surface albedo, and seasonal changes in solar cycle.

The plots and the mean values (see Table 1) suggest that the year 2019 had more cirrus coverage with OT>0.1, less
OLR and less RSR compared to 2020. The differences show a band of changes between Ireland and the Balkan
countries which resemble the expected aviation effects but are overlaid by changes from different weather. A
further simulation with the weather of 2019 and traffic of 2020 quantifies the differences coming from the changes
in weather. The mean contrail-cover in 2020, see Table 1, would have been 6 % larger if the weather in 2020 would

have been the same as in 2019. So, the weather impact on the contrail properties is smaller than the traffic impact
on contrails. Compared to the background atmosphere, the contrail induced changes reach about 10 % of the total
cirrus cover and the LW RF values reach an order 10 % of the spatial and temporal variability of OLR. The relative
contribution of SW RF to RSR is smaller because of larger variability of RSR.

**Table 1: Mean air traffic and contrail properties for traffic and weather in various years**

| Case | Unit | 1 | 2 | Ratio | 3 | Ratio |
|---|---|---|---|---|---|---|
| Traffic | | 2019 | 2020 | cases | 2020 | cases |
| Weather | | 2019 | 2019 | 2/1 | 2020 | 3/1 |
| Flight distance | Mm d$^{-1}$ | 21650 | 6110 | 28.2 % | 6110 | 28.2 % |
| Fuel consumption | Gg d$^{-1}$ | 79.69 | 22.46 | 28.2 % | 22.46 | 28.2 % |
| Flight level pressure altitude | km | 10.56 | 10.62 | 100.6 % | 10.62 | 100.6 % |
| Flight level with contrails | km | 10.78 | 10.79 | 100.1 % | 10.8 | 100.2 % |
| Flight distance with contrails | Mm d$^{-1}$ | 1626 | 501.3 | 30.8 % | 353.5 | 21.7 % |
| Contrail age | h | 2.029 | 2.073 | 102.2 % | 2.118 | 104.4 % |
| Contrail optical thickness | 1 | 0.088 | 0.100 | 114.0 % | 0.104 | 118.5 % |
| Contrail particle volume mean radius | µm | 8.65 | 8.64 | 99.8 % | 9.22 | 106.5 % |
| Contrail particle effective mean radius | µm | 14.4 | 14.5 | 100.4 % | 15.3 | 106.3 % |
| Total cirrus coverage at OT > 0.1 | 1 | 0.278 | 0.264 | 94.9 % | 0.249 | 89.5 % |
| Contrail coverage at OT > 0.1 | 1 | 0.0461 | 0.0149 | 32.4 % | 0.0140 | 30.3 % |





| | | | | | | |
|---|---|---|---|---|---|---|
| FC outgoing longwave radiation | W m$^{-2}$ | 248.4 | 248.4 | 100.0 % | 249.7 | 100.5 % |
| FC reflected shortwave radiation | W m$^{-2}$ | 114.6 | 114.6 | 100.0 % | 115 | 99.2 % |
| Longwave radiative contrail forcing | W m$^{-2}$ | 0.8992 | 0.285 | 31.7 % | 0.2668 | 29.7 % |
| Shortwave radiative contrail forcing | W m$^{-2}$ | -0.757 | -0.215 | 28.4 % | -0.2008 | 26.5 % |
| Net radiative contrail forcing | W m$^{-2}$ | 0.1422 | 0.07001 | 49.2 % | 0.066 | 46.4 % |

From plots like those shown in the lower panels of Figure 10 to Figure 12, one can read the maximum differences between 2019-2020, as listed in Table 2. The extreme values in the differences 2019-2020 are positive for OT and OLR and negative for RSR, as expected for larger contrail-cirrus cover in 2019 compared to 2020.

Comparing the values in Table 2, we note that the changes in the mean differences 2019-2020 from total cirrus and irradiances changes are 3 to 10 times larger than the changes to be expected in contrail cirrus OT and in LW and SW RF components. Obviously, weather changes had a stronger effect on these satellite-observable properties than air traffic in 2019/2020. In addition, we have to expect changes from other emissions (e.g., at the surface) not modelled in this study.

**Table 2: Extreme changes in contrail and total cirrus OT and irradiances between 2019 and 2020**

| | 2019-2020 | | 2019-2020 | Unit |
|---|---|---|---|---|
| Contrail OT | 0.054 | Total cirrus OT | 0.15 | 1 |
| LW RF | 2.2 | OLR - LW RF | 8.6 | W m$^{-2}$ |
| SW RF | -2.1 | RSR - SW RF | -20 | W m$^{-2}$ |

## 5. Parameter Studies

In addition to the variations in weather and traffic, the results are sensitive to various model and input parameters.

### 5.1 Sensitivity to the performance model used

Results from BADA3 and the new PS method (Poll and Schumann, 2020a) are very similar for fuel consumption, but there are large differences in the estimates of overall engine propulsion efficiency, η. These have consequences for the formation of contrails at threshold conditions. After preliminary studies showed that BADA3 overestimates




η, we use BADA3 η values reduced by factor 0.85 in the reference simulation in this paper. A total of 184 ICAO aircraft types (or their BADA3-synonyms) contributed to the fuel consumption over Europe in 2019 (and similar in 2020), 162 to contrails in the year 2019 and 154 in 2020. The PS model currently provides data for 54 of these aircraft types. For traffic of 2019, the PS aircraft types account for 95 % of the fleet fuel consumption and 97 % of the total contrail forcing. In 2020, their contribution to contrail forcing is 91 %. Hence, the PS model with aircraft characteristics as given in the tables of Poll and Schumann (2020a) covers 91 to 97 % of relevant aircraft types. Therefore, the PS method was used where possible and for aircraft types not covered in the current PS method and for climb and descent phases, BADA3 data are used.

As an aside, it was found that that 80 % of fuel consumption over Europe comes from just 15 aircraft types, whilst 80 % of the contrail forcing came from 13 types in 2019 and from 16 in 2020. In addition, 90 % of fuel consumption comes from 23 types, 90 % of contrail forcing comes from 19 types in 2019 and 24 in 2020. One particular aircraft type, a twin-engine medium-sized airliner, produced nearly 20 % of total fuel consumption and 16 % of contrail forcing. In 2020, the largest contrail contribution came from one type of twin-engine heavy aircraft, probably as a result of the larger fraction of cargo flights in 2020 (ICAO, 2021).

Table 3 compares results for one month's traffic (July 2019) using the original BADA3 (η not corrected by factor 0.85 as in the reference case) and PS. The integrated fuel consumption differs by less than 1 %. For individual flights, the flight-mean fuel consumption values at FL above 180 exhibit a Pearson correlation coefficient of 0.998. The η mean values and standard deviations at cruise are 0.38±0.06 for BADA3 and 0.31±0.05 for PS with relative mean difference of (20±9) % and mean correlation of 0.89. BADA3 tends to overestimate drag at cruise and, hence, engine thrust, as confirmed by a few comparisons to alternative performance models (BADA4 (Nuic et al., 2010) and PIANO (Simos, 2004)). Since contrails form at higher temperature for higher η, more contrails form in the model runs when BADA3 is used compared to when the PS model is used. As expected, the total contrail flight distances differ by only about 3 % because many contrails occur at temperatures far below the threshold temperature. The mean optical depth and the mean RF values are 3 to 5 % larger for BADA3 than for PS input. Incidentally, the net RF changes with similar magnitude, but with a different sign because the added contrails for higher η occur mainly at lower altitudes contributing more to SW than to LW forcing. This clearly illustrates the non-linearity of the climate impact of contrail formation.



**Table 3: Sensitivity to the performance models**

| Parameter | Unit | BADA3 | PS | BADA3/PS Ratio |
|---|---|---|---|---|
| Flight distance | Mm d$^{-1}$ | 24210 | 24210 | 100.0 % |
| Fuel consumption | Gg d$^{-1}$ | 87.6 | 87.62 | 100.0 % |
| Contrail distance | Mm d$^{-1}$ | 1552 | 1506 | 103.1 % |
| Mean age | h | 2.003 | 2.027 | 98.8 % |
| Contrail optical thickness | 1 | 0.1048 | 0.1003 | 104.5 % |
| Longwave RF | W m$^{-2}$ | 0.9583 | 0.933 | 102.7 % |
| Shortwave RF | W m$^{-2}$ | -0.8359 | -0.8075 | 103.5 % |
| Net RF | W m$^{-2}$ | 0.1225 | 0.1255 | 97.6 % |

**Table 4: Sensitivity to soot emission index in two CoCiP model versions**

| Model | With exchange and overlap | | | Without exchange or overlap | | |
|---|---|---|---|---|---|---|
| EI$_{soot}$/($10^{15}$ kg$^{-1}$) | 1.5 | 1 | Ratio | 1.5 | 1 | Ratio |
| Fuel burned/Gg | 87.6 | 87.6 | 100.0 % | 87.6 | 87.6 | 100.0 % |
| Distance with contrails/Mm | 1487 | 1493 | 99.6 % | 1554 | 1554 | 100.0 % |
| Mean age/h | 2.03 | 1.99 | 101.9 % | 1.98 | 1.95 | 101.6 % |
| Mean optical thickness | 0.102 | 0.083 | 122.9 % | 0.118 | 0.094 | 125.5 % |
| Volume mean radius/μm | 9.442 | 10.4 | 91.2 % | 10.6 | 11.4 | 93.1 % |
| Effective radius/μm | 15.18 | 16.5 | 92.2 % | 17.6 | 18.7 | 94.2 % |
| Longwave RF/(W m$^{-2}$) | 0.9311 | 0.788 | 118.2 % | 1.583 | 1.244 | 127.3 % |
| Shortwave RF/(W m$^{-2}$) | -0.8061 | -0.655 | 123.0 % | -1.221 | -0.937 | 130.3 % |
| Net RF/(W m$^{-2}$) | 0.125 | 0.132 | 94.3 % | 0.362 | 0.307 | 117.8 % |

## 5.2 Sensitivity to soot emissions

The soot emission indices derived with the fractal aggregate model (Teoh et al., 2019) are, even after multiplication with the above-mentioned adjustment factor 0.5, on average 50 % larger than the fixed value $1\times10^{15}$ kg$^{-1}$ used in an earlier CoCiP study (Schumann et al., 2015). As expected (Teoh et al., 2020b), Table 4 shows that a 50 % larger



soot emission index causes a slightly larger contrail age (2 %), larger optical contrail thickness (25 %) and 20 to

30 % larger RF values, with largest impact on the SW part. The increased particle number enhances SW effects more than LW. That is a known phenomenon, see figure 10 in Schumann et al. (2012).

## 5.3 Importance of relative humidity

**Table 5: Sensitivity to mean ice supersaturation parameter RHic, absolute values and ratios relative to the reference case 2.**

| Case | 1 | 2 | 3 | Ratios | |
|---|---|---|---|---|---|
| RHic | 1 | 0.95 | 0.9 | 1 to 2 | 3 to 2 |
| Contrail distance/Mm | 807.4 | 1487 | 2071 | 54 % | 139 % |
| Mean age/h | 2.04 | 2.03 | 2.05 | 100 % | 101 % |
| Mean optical thickness | 0.0867 | 0.102 | 0.124 | 85 % | 122 % |
| Longwave RF | 0.434 | 0.931 | 1.644 | 47 % | 177 % |
| Shortwave RF | -0.372 | -0.806 | -1.439 | 46 % | 179 % |
| Net RF | 0.061 | 0.125 | 0.205 | 49 % | 164 % |


The amount of ice supersaturation in the background atmosphere is the most important parameter for contrail modelling. The inverse of the parameter RHic is used to enhance humidity. Table 5 shows the sensitivity of domain mean values for one month with dense traffic (July 2019) to changes in RHic. Both absolute and relative values are given, compared to the results for RHic = 0.95. As expected, both the contrail length (flight distance with contrail

formation) and their optical thickness increase strongly with increasing humidity. The overall impact of increasing, or decreasing, RHic by 5 % are changes in net RF of order 60 %. Obviously, the sensitivity to RHic is significant and the RHic value selected should be checked again when comparing the model results to observations.

Several other parameters are also important. For example, enhancing the vertical shear of horizontal wind by factor of 2, or vertical diffusivities by similar amounts causes changes in RF of order 10 to 20 %.

**5.4 Sensitivity to the water vapor exchange and contrail overlap model**

As can be seen from Table 6, the water exchange reduces the contrail optical thickness and the RF values by 10 to 20 %, with the larger values being for the denser traffic in 2019. The water exchange causes less ice particle





sedimentation and, hence, increases contrail lifetime on average by 1 to 4 %, with the larger values for 2019 traffic. This is consistent with the results from a study with CoCiP coupled to a climate model (Schumann et al., 2015).

**Table 6**: Effects of water exchange and contrail overlap

| Traffic | 2019 | 2019 | 2019 | 2020 | 2020 | 2020 |
|---|---|---|---|---|---|---|
| Water exchange | no | yes | yes | no | yes | yes |
| Overlap | no | no | yes | no | no | yes |
| Flight distance/Mm | 24210 | 24210 | 24210 | 8202 | 8202 | 8202 |
| Fuel mass burned/Gg | 87.6 | 87.6 | 87.6 | 26.7 | 26.7 | 26.7 |
| Contrail age/h | 1.98 | 2.02 | 2.03 | 2.00 | 2.01 | 2.02 |
| Optical thickness | 0.12 | 0.10 | 0.10 | 0.12 | 0.11 | 0.11 |
| Volume radius/µm | 10.6 | 9.6 | 9.4 | 10.3 | 9.7 | 9.6 |
| Effective radius/µm | 17.6 | 15.4 | 15.2 | 17.0 | 15.6 | 15.5 |
| Longwave RF/(W m$^{-2}$) | 1.583 | 1.280 | 0.931 | 0.429 | 0.384 | 0.329 |
| Shortwave RF/(W m$^{-2}$) | -1.221 | -0.993 | -0.806 | -0.322 | -0.289 | -0.260 |
| Net RF/(W m$^{-2}$) | 0.362 | 0.288 | 0.125 | 0.107 | 0.095 | 0.070 |
| *Ratios of RF values* | | | | | | |
| Longwave RF | 100 % | 81 % | 59 % | 100 % | 89 % | 77 % |
| Shortwave RF | 100 % | 81 % | 66 % | 100 % | 90 % | 81 % |
| Net RF | 100 % | 79 % | 35 % | 100 % | 88 % | 65 % |

The contrail-contrail overlap causes a significant reduction in RF. In particular, the mean LW RF is reduced by 23 % for 2020 and by 41 % for 2019. The smaller reduction of the SW RF causes up to 65 % reduction in the net RF. Hence, these overlap aspects are important when considering regions with high traffic density. The changes appear

to be larger than expected (Sanz-Morère et al., 2020).

## 6. Conclusions and Outlook

In connection with the COVID-19 pandemic, global air traffic was considerably less in 2020 with respect to 2019 levels. This study has quantified air traffic and contrail changes within a European dense traffic area (20°W-20°E, 35°N-60°N), from March to August 2020, compared to same months in 2019, using traffic data, emission estimates,





ECMWF weather data and a contrail model. The traffic data show that total flight-distance (in respect to air) in the
European investigation domain for traffic operating above FL 180 was 72 % smaller in 2020 than in 2019. The
changes in the total fuel consumption and soot emissions are similar. In the reference case, the model shows that
the flight distance with persistent contrail formation was reduced even more strongly, by 78 %, mainly because the
weather conditions in 2020 were less favourable for contrail formation than in 2019. The coverage of contrails with

an optical depth larger than 0.1 decreased from 4.6 % in 2019 to 1.4 % in 2020. These are large changes in view of
the about 25 to 28 % mean cirrus cover. The reduced contrail coverage caused 70 % less LW and 73 % less SW
RF with the consequential reduction of 54 % in net RF.

In order to cover flights contributing to contrail formation as completely as possible, traffic data have been derived
from a number of sources. There may still be gaps, or inaccuracies, over the Atlantic, where flight plan data have

been used. This is particularly true south of the Shanwick area and, possibly, further north and over Ireland where
detailed traffic data are missing. In all other areas the traffic should be accurately covered. The fuel consumption
is assessed using two performance models, BADA3 and the new PS, and the results are similar. In estimating fuel
use, the main uncertainty results from the unknown aircraft take-off mass. In this study, the take-off mass is
determined by using the aircraft characteristics and an assumed mass load factor, i.e. payload mass fraction of

maximum permitted payload. There are indications that the load factor was considerably reduced in the 2020
COVID-19 period. The new performance model PS provides a more accurate aircraft drag estimate at cruise giving
a 10 to 30 % reduction in the engine overall propulsion efficiency compared to BADA3. This affects contrails
under threshold conditions and reduces contrail cover by about 3 % in total. As shown recently (Teoh et al., 2020b),
the soot number emissions are larger than assumed in early contrail studies. A 50 % increase in the soot number

results in a 30 % higher net RF. This again shows the importance of soot emissions and related fuel properties
(Moore et al., 2017).

The contrail model includes a new, approximate method to account for water vapor exchange between contrails
and background air and for RF in case of contrail-contrail overlap. Water vapor exchange reduces the modelled RF
magnitudes by about 10 to 20 %, with larger values being for the denser traffic in 2019. The contrail-contrail

overlap has an even stronger effect because the irradiances depend on the area covered by contrails, while the
amount of water vapor exchange depends on the contrail volume and the volume fraction per grid cell of the rather
thin contrails is smaller than their area fraction. The 2020/2019 reductions in LW RF are larger than in SW RF
causing a smaller reduction in net RF.

It may not be easy to identify air-traffic induced changes in cirrus and irradiances over Europe in observations. The changes in total cirrus cover and irradiance values due to aviation are below 10 % of the background cirrus cover and the TOA irradiances without air traffic, in particular for SW irradiances. The aviation induced changes are 3 to 10 times smaller than the mean differences in total cirrus and in TOA irradiances caused by weather changes in 2019-2020. These ratios are sensitive to model uncertainties. The 2019-2020 changes in weather may have larger effects on contrail cirrus and its RF than the large traffic changes during COVID-19. Changes may also be caused by other aircraft emission (e.g., nitrogen oxides) (Brasseur et al., 2016) and by surface emissions.

Still, the traffic changes are large and last longer than the six months investigated so far. The traffic and the background atmosphere appear to be well characterized, and the contrail model has proven skill as demonstrated here again by comparison to a set of contrail observations. Much of the weather impact on background cirrus and irradiance changes 2019-2020 is described by the IFS weather model. A 10 % change in cirrus cover and 10 % changes in OLR relative to the regional and temporal variability are not small, and regional and diurnal variation patterns may be detectable in observations. This may allow for the detection of aviation-induced changes in that region. It will be interesting to test this hypothesis.

### Code and Data availability

The contrail model input and output data are available on request and will be made accessible in a public data repository, see Supplement. The contrail model code can be made available by the lead author on request.

### Author contributions

US performed the study and wrote the manuscript. IP contributed to performance modelling, RT contributed to contrail and soot modelling and preprocessed NATS data, RK, ES, JM and GSK contributed traffic data and related know how, RB prepared the ECMWF data, LB provided input for preparing comparisons to satellite data, MS and CV contributed to the conceptual design and many details of the study. All authors contributed to manuscript editing.

### Competing interests

The authors declare no competing interests.





## Figures

**Figure 1: Geographic map of the European domain under consideration for contrail simulations with coloured flight tracks for two half-hour example time periods of 1 March 2020, before the COVID-19 crisis. Individual panels show track data from a) CPR (red), b) M3 (blue), c) FR24 (green) and d) from the combination of CPR with M3 and NATS data for flights extending beyond the CPR range (black, blue, and purple lines).**


**Figure 2: Mean values of a) flight distance in air, b) ratio of mean flight distance with contrails to total flight distance, c) longwave (LW) RF and d) net RF versus time from 1 March to 30 August in 2019 (black curves) and 2020 (red). The data represent averages over the European domain and over a 24-h day.**





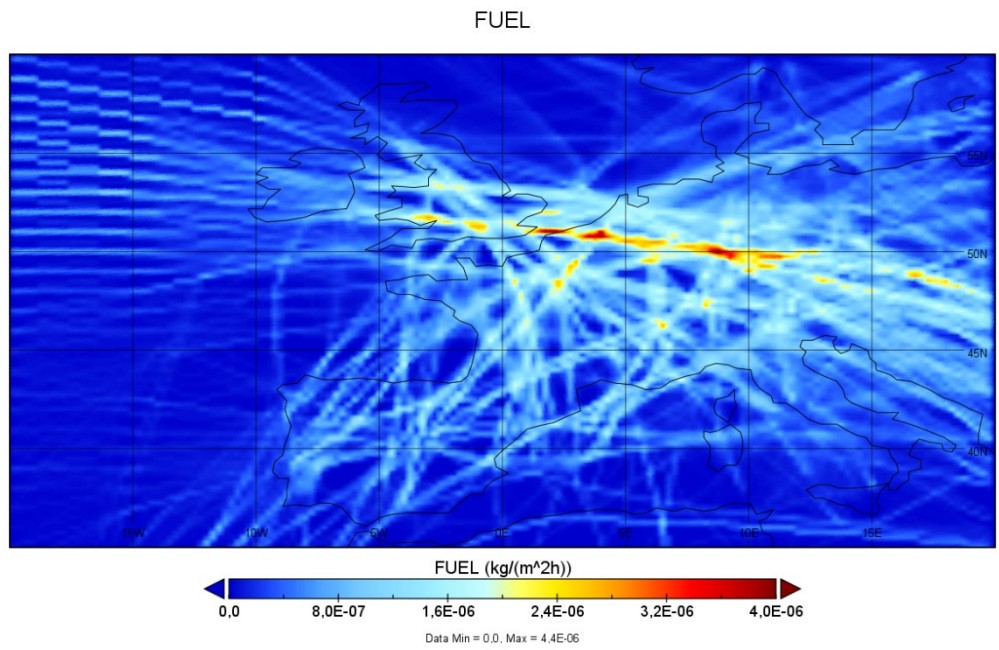

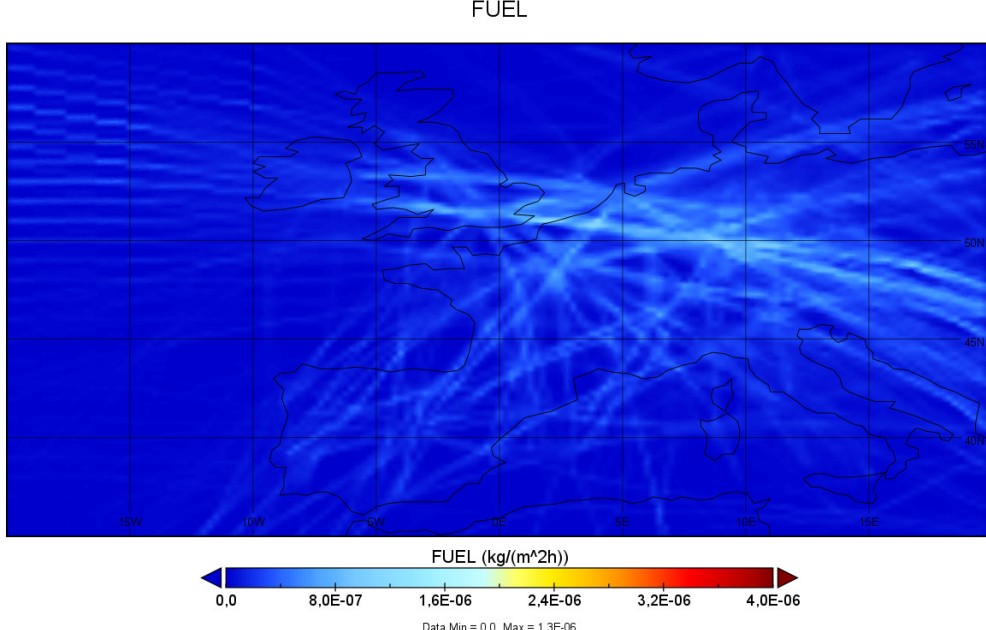


**Figure 3: Mean fuel consumption (in kg m⁻² h⁻¹) over the European domain, March-August mean values, 2019 (top) and 2020 (bottom).**



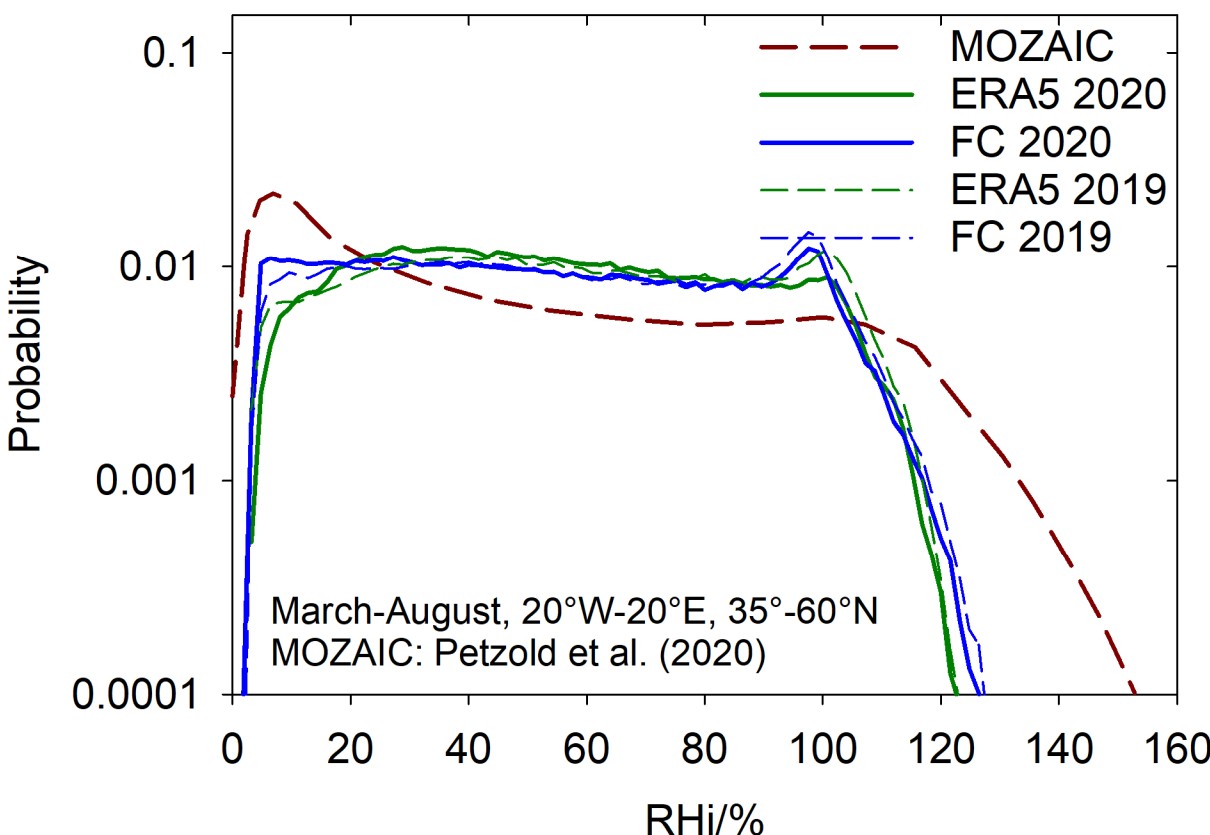


**Figure 4: Probability density of relative humidity over ice (RHi) from ECMWF IFS forecast data (FC, blue lines) and ERA5 re-analysis data (green) along the traffic routes over Europe as in 2020, separately for meteorology of 2019 and 2020. The dark red dashed curve represents the 1995-2010 MOZAIC data as in figure 5a of Petzold et al. (2020).**





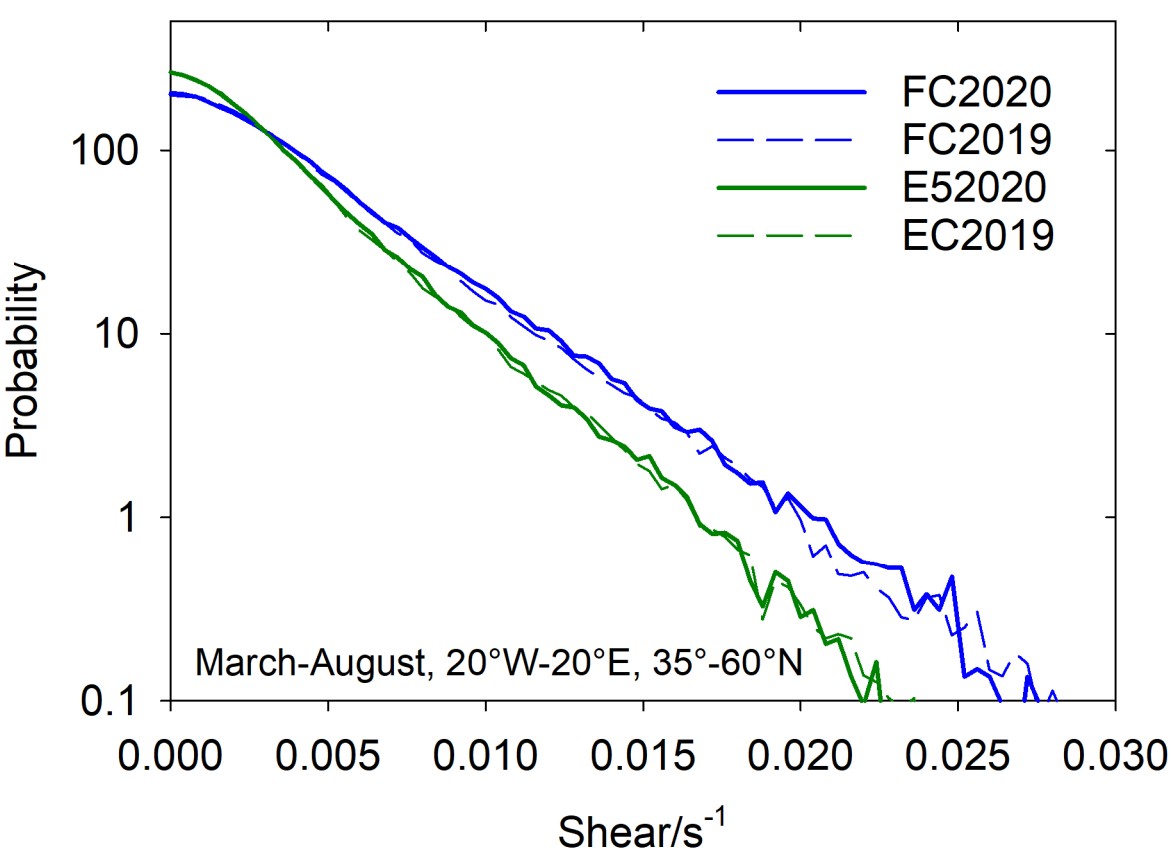

**Figure 5: Probability density of vertical shear of horizontal wind normal to flight segments along the traffic routes over Europe as in 2020, separately for FC and ERA5 meteorology as in Figure 4.**





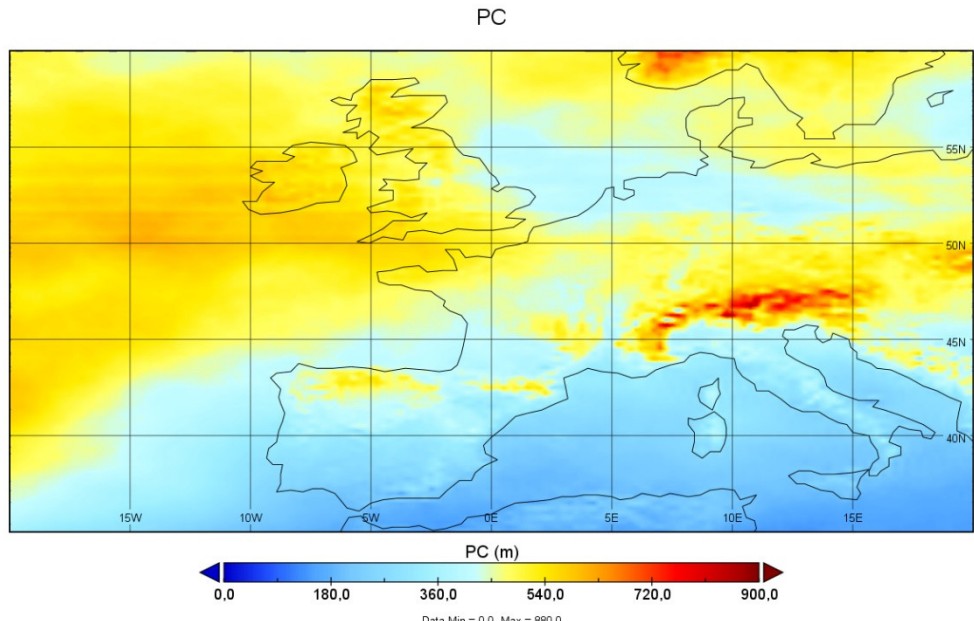

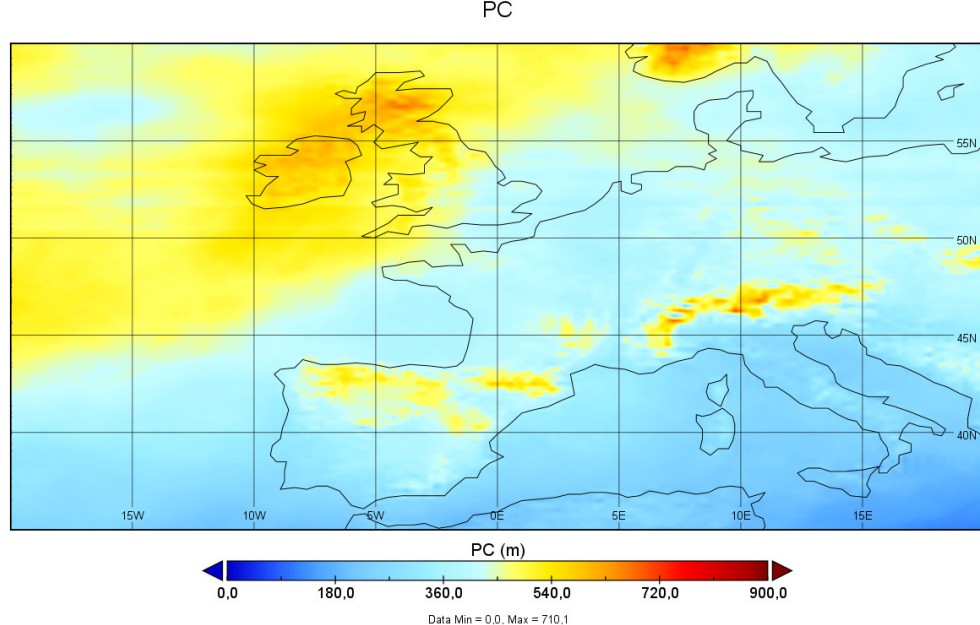


**Figure 6: Mean vertical thickness (in m) of layers conditioned for formation of persistent contrails in March-August 2019 (top) and 2020 (bottom).**





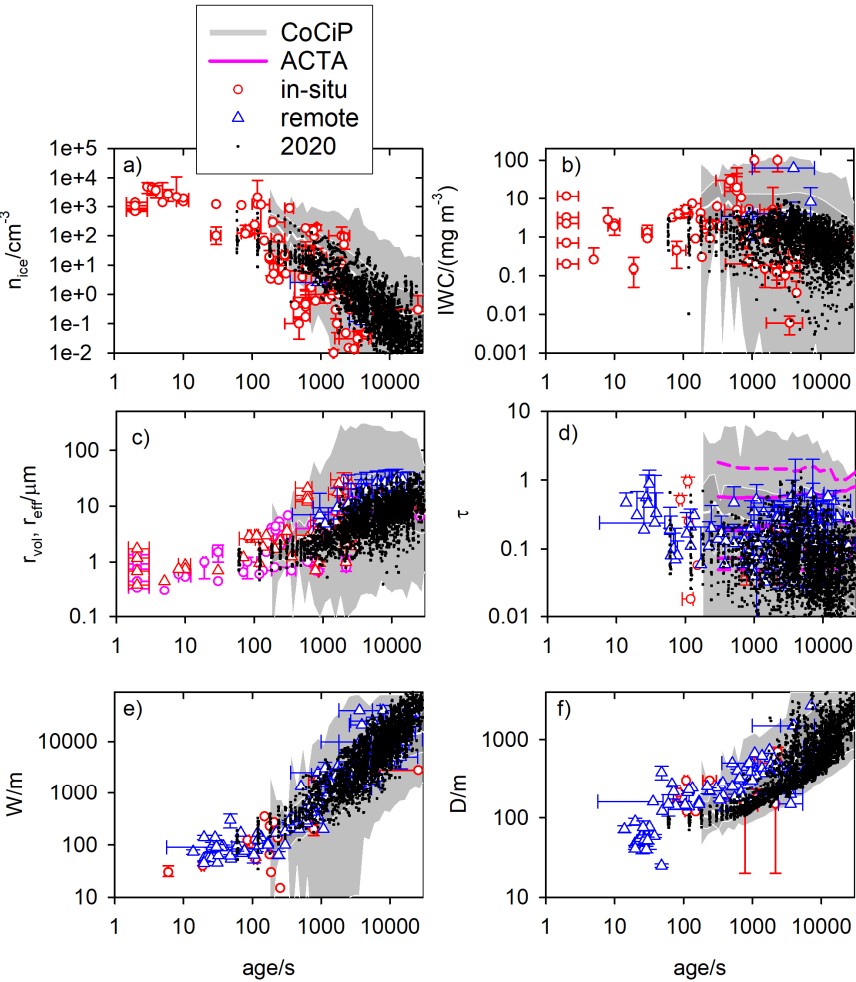

**Figure 7: Comparison of contrail model results with observed contrail properties versus contrail age. The grey areas with white lines representing 0, 10, 50, 90 and 100 % percentiles are from earlier multi-year CoCiP model results (Schumann et al., 2015). The coloured symbols denote observations from in situ and remote sensing measurements. The panels show (a) ice particle number concentration $n_{ice}$, (b) ice water content IWC, (c) volume mean and effective ice particle radius $r_{vol}$ and $r_{eff}$, (d) optical thickness $\tau$, (e) geometrical contrail width $W$, and (f) total geometrical contrail depth $D$. The purple lines in panel (d) are derived with the Automatic Contrail Detection Algorithm (ACTA) algorithm from satellite observations (Vázquez-Navarro et al., 2015). The black symbols which are overlaid over this previously published figure (Schumann et al., 2017; Schumann and Heymsfield, 2017) show computed contrail properties from the present study for a random subset of flight segments from 2020 in the reference model version.**





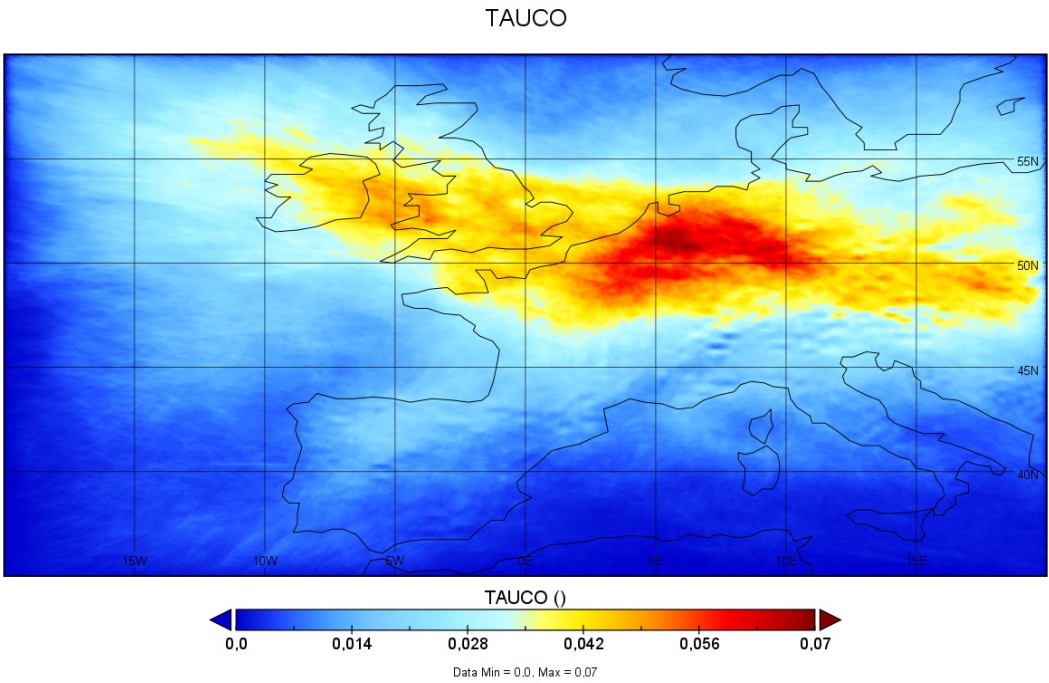

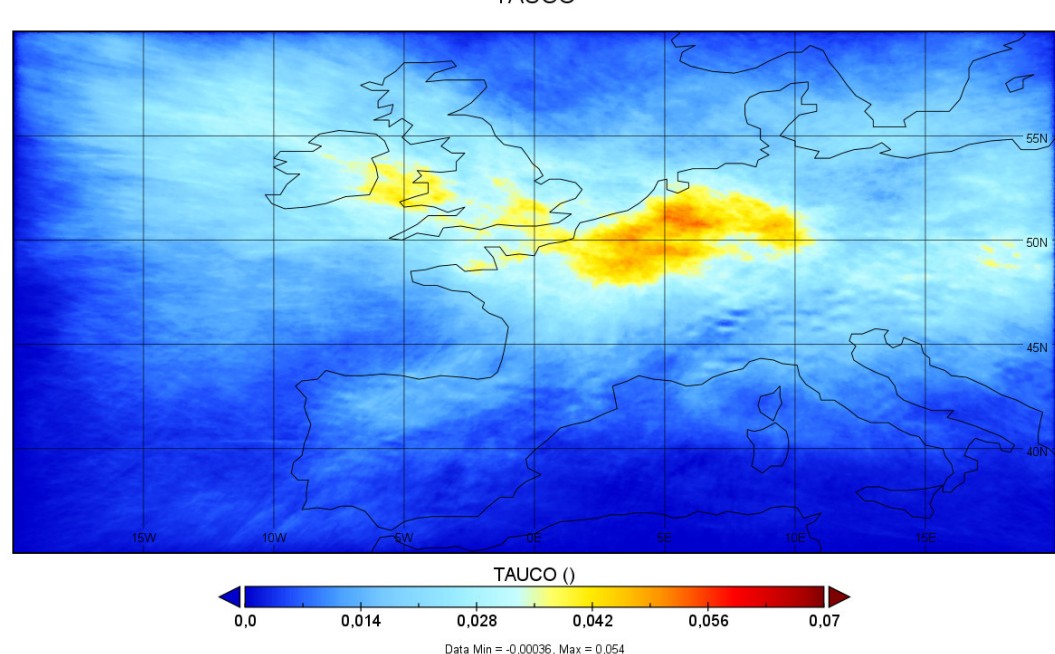

**Figure 8: Mean optical thickness of contrails, March-August mean, 2019 (top) and difference 2019-2020**
**(bottom).**





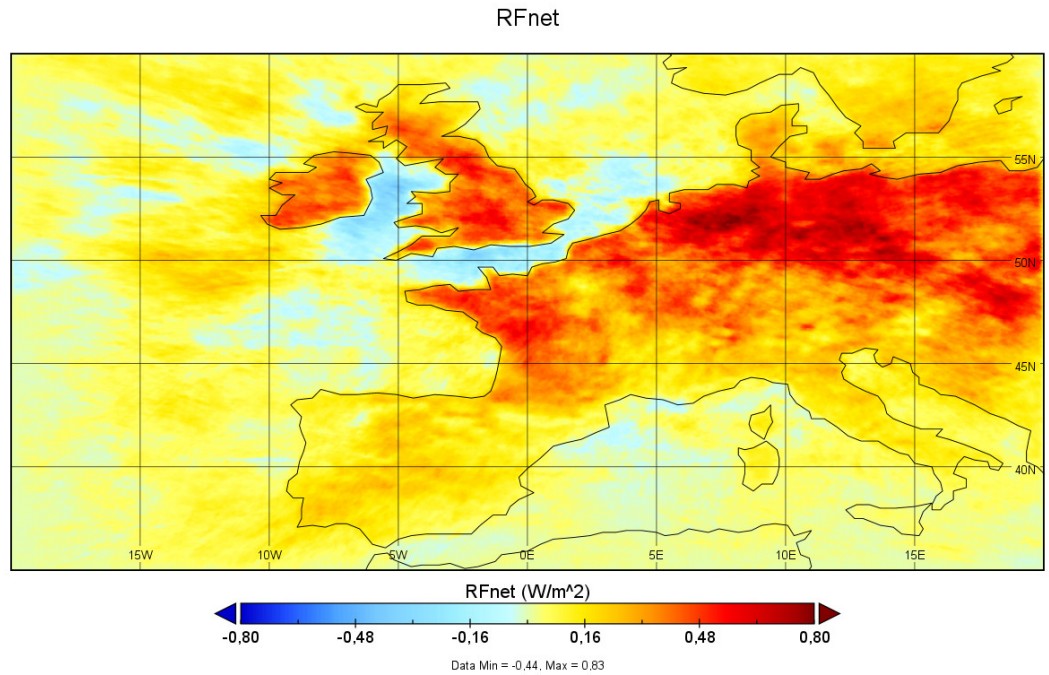

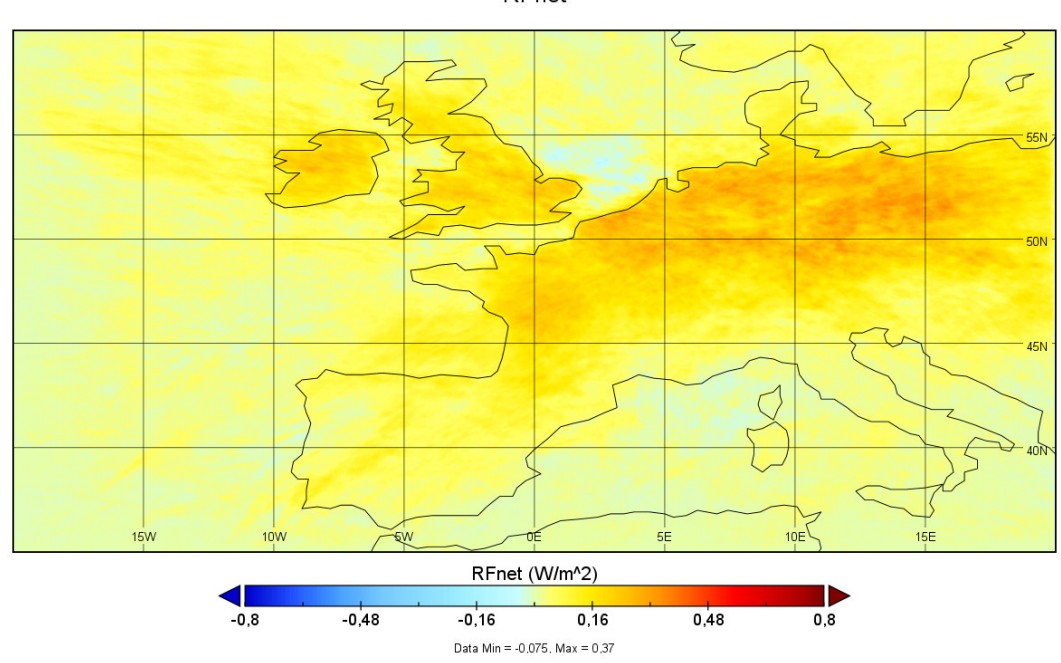

**Figure 9: Mean net RF in W m⁻² from contrails, March-August mean, 2019 (top) and 2020 (bottom).**





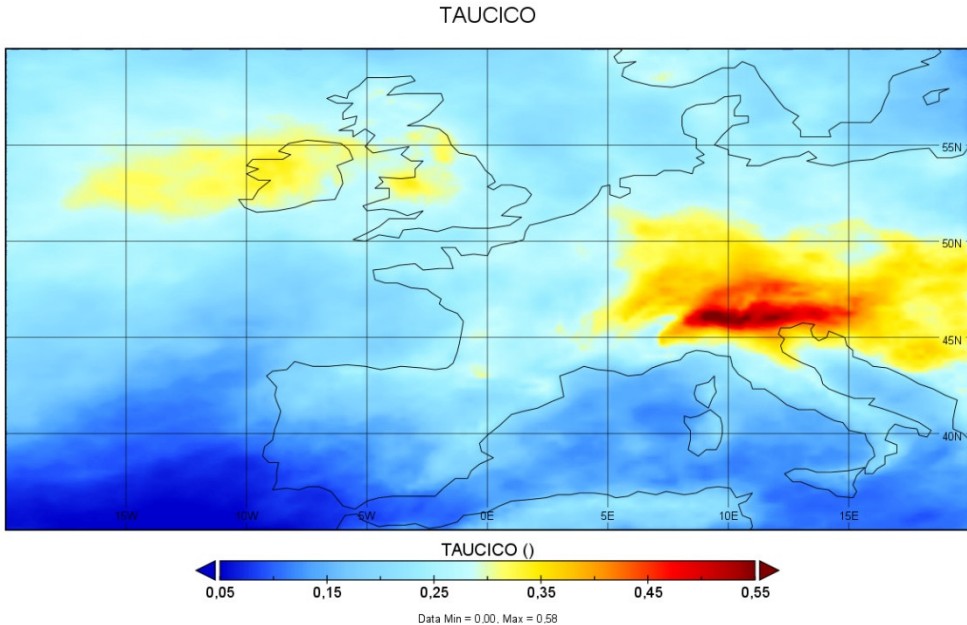


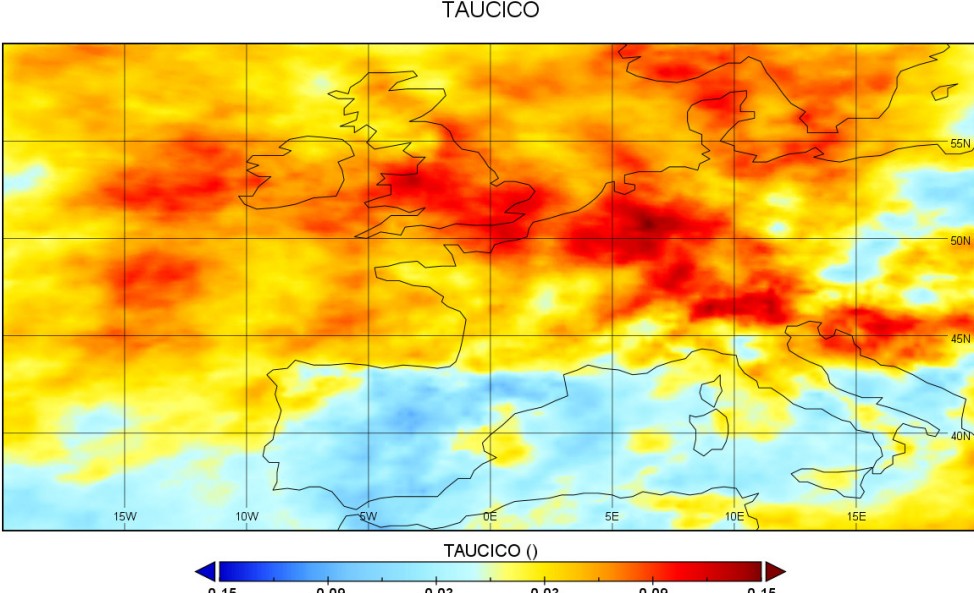

**Figure 10: Mean cirrus optical thickness (OT) (nondimensional) in the sum of IFS and CoCiP results, March-August mean, 2019 (top) and difference 2019-2020 (bottom).**





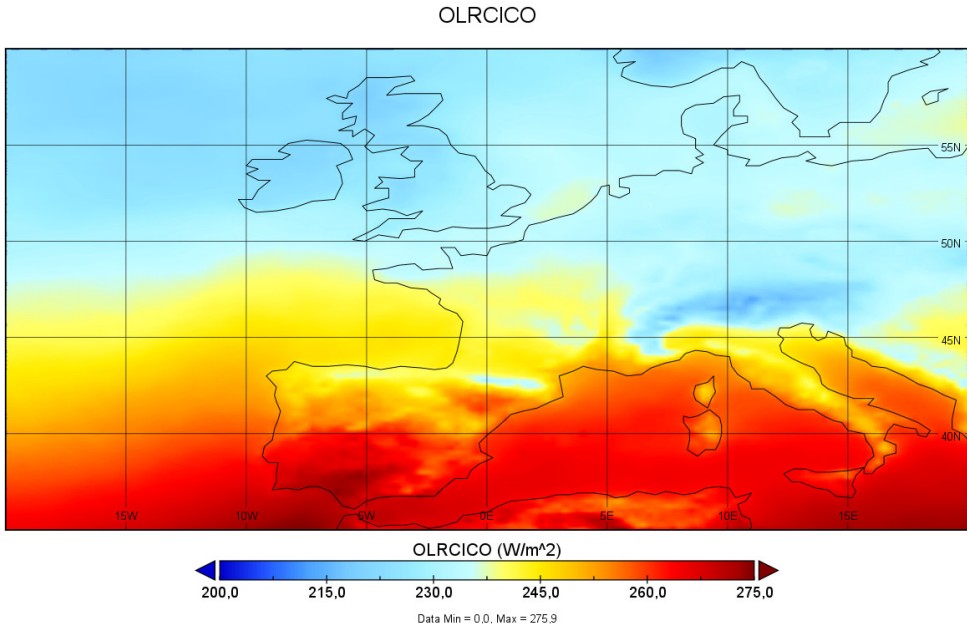


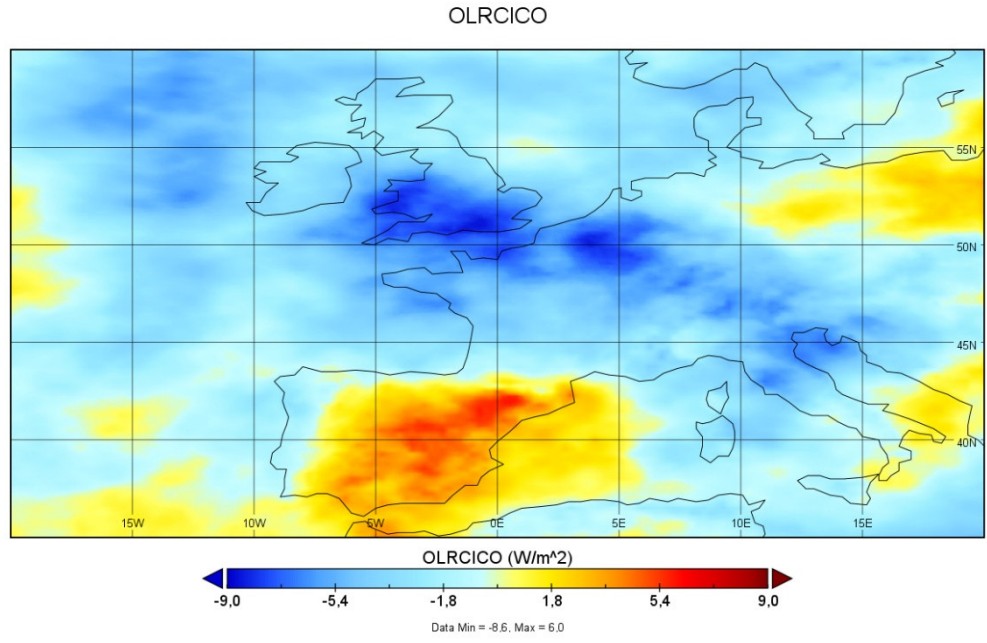

**Figure 11: Mean Outgoing Longwave Radiation (OLR) in W m⁻² in the sum of IFS and CoCiP results, March-August mean, 2019 (top) and difference 2019-2020 (bottom).**



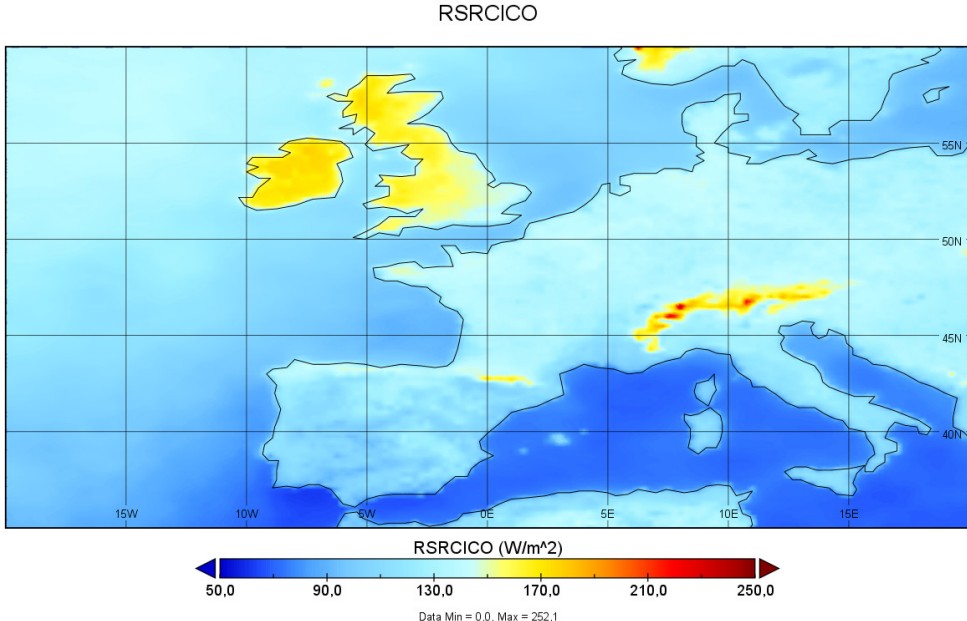


Figure 12: Mean Reflected Solar Radiation (RSR) in W m⁻² in the sum of IFS and CoCiP results, March-August mean, 2019 (top) and difference 2019-2020 (bottom).




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
