# Peer review of "Air traffic and contrail changes during COVID-19 over Europe: A model study"

_Atmospheric Chemistry and Physics, 2021_

## Author Response (AR1)

**Reply to comments by Referee #1.**

Subsequently, the Referee's comments are repeated in blue color. Our replies follow in black.

We thank the Referee for his (or her) careful review. We respond to the recommendations by extra analyses and by added discussions as explained below.

***Summary***
The authors use a contrail model to quantify the change in contrail cover over Europe between 2019 and 2020, and the sensitivity of simulated contrail coverage to different modeling assumptions. They find that contrail coverage over Europe fell substantially over Europe during 2020, but that the overall change in net RF was smaller than the change in
total flight distance. They gain some insight into the root cause of this through a counterfactual analysis, using mismatched weather and traffic data to quantify the role that meteorological variability had in the difference. They also find that implementation of new contrail interaction terms could reduce the simulated net radiative forcing due to contrails by up to 65% - an intriguing result.

We agree.

      The central question of the manuscript is the degree to which contrail coverage over Europe declined between 2019 and 2020, and whether this was proportional to the reduction in flight distance. This question is interesting, but without comparisons to
observations its accuracy cannot be evaluated. The authors tease such a comparison but defer it entirely to a second paper.

The second paper, comparing the model results with satellite observations, just got accepted for publication. We refer to this at various places in the revised paper.

Schumann, U., L. Bugliaro, A. Dörnbrack, R. Baumann, and C. Voigt: Aviation contrail cirrus and radiative forcing over
Europe during six months of COVID-19, Geophys. Res. Lett., doi: 10.1029/2021GL092771, 2021.

Ideally one should have accurate and representative observations which allow to assess the accuracy of the model predictions. However, when we started this study, such observations were not available. Even now, with some recent observations, the accuracy of model predictions can only be estimated because the observations have their own limitations.

In the absence of observations, it is common practice to present the results from model variants and from parameter studies to get some insight into the possible range of model results. The study shows that the range of model results is large so that a final estimate of contrail effects requires a careful combination of model and observation results.

The authors are also unable to address the question of why this mismatch was present, beyond showing that weather variability alone is not sufficient to explain it. As such, while the data do support the conclusions, and the methods are appropriate for the relatively narrow scope of the question, the paper constitutes only an incremental advance. In addition, roughly half of the results section is
taken up with discussion of methodological advances which seem to have little to do with the central question of the paper.

Overall, it is useful to have the first data point for the likely change in contrail coverage resulting from air traffic reductions in 2020, and the technical advances made in the CoCiP model are significant.

However, the paper would be substantially improved by the inclusion of a comparison to observations, and by either moving the discussion of methodological advances into a separate paper or refocusing their analysis on the central question of the paper. The former in particular would raise the scientific significance of the paper and make the title more appropriate.

We thank the reviewer for this supportive comment. The more critical parts will be reflected in the discussion as also provided below.

*Major comments*
The overall goal of the paper seems confused. From the abstract alone, two separate
objectives are clear: to quantify the change in contrail cover during 2020 compared to 2019 (lines 15-26), and to determine the sensitivity of the CoCiP model to certain model parameters (lines 26-32). Similarly, I count roughly four pages of discussion of the differences in simulated contrails between 2019 and 2020, and around the same number of pages of discussion of the effect of model parameters on simulated coverage. The
problem is that these two disparate components do not add up to a complete study, in part because there seems to be little meaningful overlap between the two. Given that the title of the paper is specifically focused on the effect of COVID-19 on air traffic and contrail coverage, my recommendation would be to focus the discussion in section 5 on the question of whether these model advances affect the conclusions of the paper, rather than
the current more abstract discussion of the effect they have on individual years.

We agree that this study cannot yet cover all aspects of the complex problem. Further studies are to come. This is reflected in the text and the final sentences of the Conclusions.

Specifically, to what degree does each of these model advances change the effect that
changes in air traffic had on contrail coverage in 2020 compared to 2019? As it stands, the discussion in section 5 almost exclusively discusses what each of these advances does to the estimated contrail coverage for one year at a time without covering the implications for our understanding of the changes between 2019 and 2020.

The model changes show indeed quantitative differences. However, the qualitative results are unchanged
by the parameters varied. See the conclusions.

Such an analysis would also illuminate the most important observation in the paper, which is (lines 319-321) that the reduction in the net RF was smaller than the reduction in traffic. Currently, the only explanation offered is that "[t]his is due, in part, to different
changes of SW and LW RF and to the nonlinear effects from contrail-background humidity exchanges and contrail-contrail overlap". The paper would benefit from a rearrangement of the analysis to focus on why the traffic, coverage, and RF changes were not all proportional. While this is partially addressed in the paper, the message of which factors contribute what (other than the separation of meteorological factors) is not communicated
clearly.

For applications it is certainly important to note that the net RF may change different from the SW and LW parts. However, this should not be surprising. The SW and LW RF values have opposing sign and the SW and LW magnitudes are about a factor of 4 to 6 larger than the net RF magnitude. Hence, small changes in the RF components have large impact on the net RF. We note that the SW and LW RF

components depend both on the contrail cirrus optical depth, and are correlated, therefore. Thick contrails cause both large LW and SW RF values. However, the correlation is far different from 100 % because the SW and LW effects depend on different input values (temperature, solar zenith angle, particle habits. particle sizes, system albedo, incoming solar irradiance and outgoing longwave irradiance) and respond to changes in the input parameters with different sensitivities. The correlations are zero during night, of course, and the day/night duration ratio and other input parameters are variable over the time period considered.  Also, the day/night traffic ratio varied between the years. Therefore, several reasons caused different relative changes of net RF compared to the LW and SW components. We do not see a reason why the change in net RF should be always smaller than the change in the two components. This behavior may be peculiar to the situation considered.

This is now discussed, among others,  in the conclusions

My final major comment is that the paper seems like it would be best served by separation into two parts. The authors mention several times that they are in the process of developing a follow-on paper which will compare the model results to satellite data. It would appear much cleaner if the current manuscript were focused specifically on the model advances, rather than on the effect of COVID on contrail coverage. This would allow the aforementioned second paper to cleanly introduce both the model-based estimate of changes in contrail coverage due to COVID in a context where the results could be validated against observations. Naturally this is at the discretion of the authors, but such a division would resolve many of the concerns I have above.

We approach the problem stepwise. This paper describes the traffic and the contrail modelling. A companion paper (submitted) describes the results of a comparison with observations for 6 months. This is not the end of this line of research. Further studies are needed to explain the differences between the model and observation results because the observed changes are caused not only by contrails but also by other anthropogenic and natural effects. So this paper opens a new approach to study the climate impact of aviation and other anthropogenic changes during the COVID-19 pandemic..

*Minor comments*

Line 96: "Piston engine power aircraft only make a very small contribution" – a citation or quantification is needed for this.

We quantify the fraction of various aircraft types below.

Line 190: "The decrease of fuel consumption and flight distances are similar because the relative increase in aircraft weight (more cargo aircraft) is largely balanced by the lower load factor". Can you provide some quantification or reference? This is an interesting observation and potentially relevant to the discussion.

We quantify the aircraft mass below.

Line 420: The statement "The changes appear to be larger than expected" seems incorrect. Given the larger optical depth and narrower regional scope of this paper, the two studies seem to be in broad agreement – noting Sanz-Morère's discussion of the increase in overlap RF effects with optical depth.

We agree and changed the paper accordingly.

New: As predicted (Sanz-Morère et al., 2021), these overlap aspects are important for regions with high traffic density.

**Traffic changes for different aircraft sizes and types**

In response to the questions raised by Reviewer #1, we provide more detailed information on the changes in aircraft types and aircraft masses during the COVID-19 period. This information will be included in the revised version of the paper (either in the main text or in the supplement).

As a result of the sudden change in demand and permissions for air transport, fleet operations in 2020 were very different from 2019.

Table 1 compares total air distance covered in flights above FL180 over the European domain in March-August 2020 compared to March-August 2019. Here, aircraft are split into 5 mass classes, as explained in the table caption, depending on the maximum permitted take-off mass (MTOM), using BADA3 data for given ICAO aircraft types.

In April 2020, the total distance flown decreased to 8.8 % of the April 2019 values. The reduction was strongest for light and medium sized aircraft, i.e. single aisle transport and business jets, whilst general aviation aircraft (< 20 Mg) and heavy aircraft, i.e. twin aisle transport and cargo, experienced smaller reductions. By July 2020, light aircraft flight distances had returned to 70 % compared to the year before, whilst the average overall reduction was 23 % compared to July 2019.

**Table 1.** Flight distances (in Gm) of general aviation/military jets (G: MTOM < 20 Mg), light (L: 20 < MTOM <= 46 Mg), medium (M: 46 < MTOM/Mg <= 115), heavy (H: 126 < MTOM/Mg <= 395 Mg) and super heavy (S: 395 < MTOM/Mg) aircraft over Europe above FL 180, in the months April (4) and July (7), in 2019 and 2020; absolute values and percentage fractions of 2019 values.

| Year | Month | G | L | M | H | S | Total |
|------|-------|---|---|---|---|---|-------|
| Absolute values | | | | | | | |
| 2019 | 4 | 0.69 | 1.64 | 31.25 | 10.58 | 1.56 | 45.72 |
| 2020 | 4 | 0.16 | 0.12 | 0.82 | 2.37 | 0.51 | 3.98 |
| 2019 | 7 | 0.55 | 1.25 | 36.40 | 10.63 | 1.48 | 50.32 |
| 2020 | 7 | 0.39 | 0.35 | 7.42 | 3.04 | 0.49 | 11.68 |
| Relative values | | | | | | | |
| 2019 | 4 | 100.0% | 100.0% | 100.0% | 100.0% | 100.0% | 100.0% |
| 2020 | 4 | 22.8% | 7.5% | 2.6% | 22.4% | 32.6% | 8.7% |
| 2019 | 7 | 100.0% | 100.0% | 100.0% | 100.0% | 100.0% | 100.0% |
| 2020 | 7 | 70.6% | 27.8% | 20.4% | 28.6% | 32.8% | 23.2% |

Turbofan powered (jet) aircraft are responsible for most of the air distance flown at Flight Levels above 180 (>97.6 %) and for >99.6 % of all contrails, see Table 2. The contribution to air distance flown from turboprops is far smaller (<3.11 %) and even less for contrails (<0.36 %). The contrail contribution from piston-engine aircraft is below 0.05%, largely because they usually operate at altitudes below FL180.

**Table 2.** Total flight air distances and flight air distances with persistent contrails (in Gm) of jet, turboprop and piston-engine aircraft over Europe above FL 180, in the months April and July, in 2019 and 2020; absolute values and percentage fractions of monthly totals.

| Year | Month | Jet | Turboprop | Piston | Total | Jet | Turboprop | Piston | Total |
|------|-------|-----|-----------|--------|-------|-----|-----------|--------|-------|
| Flight distance | | | | | | | | | |
| 2019 | 4 | 592.05 | 8.976 | 0.066 | 601.1 | 98.5% | 1.49% | 0.011% | 100% |
| 2020 | 4 | 51.31 | 1.648 | 0.040 | 53.0 | 96.8% | 3.11% | 0.076% | 100% |
| 2019 | 7 | 739.24 | 9.660 | 0.212 | 749.1 | 98.7% | 1.29% | 0.028% | 100% |
| 2020 | 7 | 247.92 | 5.957 | 0.105 | 254.0 | 97.6% | 2.35% | 0.041% | 100% |
| Contrail length | | | | | | | | | |
| 2019 | 4 | 45.59 | 0.119 | 0.001 | 45.71 | 99.7% | 0.26% | 0.003% | 100% |
| 2020 | 4 | 3.96 | 0.014 | 0.002 | 3.98 | 99.6% | 0.36% | 0.046% | 100% |
| 2019 | 7 | 50.29 | 0.018 | 0.008 | 50.31 | 99.9% | 0.04% | 0.016% | 100% |
| 2020 | 7 | 11.67 | 0.015 | 0.001 | 11.68 | 99.9% | 0.12% | 0.006% | 100% |

**Reply to comments by Referee #2.**

Subsequently, the Referee's comments are repeated in blue color. Our replies follow in black.

We thank the Referee for his (or her) positive review and the discussion on ICAO CAEP's new nvPM mass and number emissions information. We respond to the remarks as explained below.

This is an excellent and timely study of the climatic impact of contrails. The authors have published extensively in this technical area and are using a tool that has been well exercised in studying contrail impact. This study has addressed a key issue and taken advantage of an unfortunate, timely reduction in air traffic due to the COVID19 pandemic to perform a (somewhat) controlled experiment to determine the radiative effects of contrails. Such a specific change is a key climatic impact is rare, and while annual changes in weather must be, and have, been taken into account, this event provides a unique opportunity to try to quantify this particular impact, largely in isolation. The authors are to be commended for noting this opportunity and taking steps to acquire and process the data to evaluate the climatic impact of contrails.

At the same time, the tool has been refined and evaluated in a few key ways to further develop and improve the model (water vapor exchange between contrails and background air, and accounting for contrail overlap). These updates have been applied to both the before and after COVID19 cases, so direct comparisons are appropriate. These are useful extensions to the modeling approach.

Thus, the paper is very scientifically interesting and offers timely analysis of the aviation climatic impact, as the industry plans recovery from a significant reduction in commercial activity. The paper is well-written and clearly presents the approach and the conclusions.

Reply: We thank the reviewer for this assessment.

I only have a few comments that I hope will improve the clarity of the excellent disposition of this useful analysis.

Lines 162 et seq.: The analysis makes use of the ICAO emissions databank to obtain soot emissions indices. I presume that they performed this analysis prior to the publication of the new nvPM entries in the ICAO Edb, which were released in December 2020. Thus, they presumably used the earlier ICAO Edb entries for Smoke Number (SN) to estimate soot parameters. Given that the bulk of the work was done months before the nvPM ICAO data was released, they are unlikely to have been able to use the new nvPM data. However, for readers that are reviewing these results now and later, when the nvPM ICAO data is now available, it is probably important to point out explicitly that they have made their soot parameter estimation based on SN data in the ICAO data bank.

Reply: Thank you for this important question. It helps us to clarify the method used:

The black carbon (BC) number emissions index ($EI_n$) is calculated using the Fractal Aggregates (FA) model (Teoh et al., 2020): it estimates the BC $EI_n$ from the BC mass emissions index (BC $EI_m$), particle size distribution (geometric mean diameter, GMD, and its standard deviation, GSD) and morphology ($D_{fm}$):

For each flight, the BC $EI_m$ is estimated using the Formation and Oxidization Method (FOX) (Stettler et al., 2013) and Improved FOX method (ImFOX) (Abrahamson et al., 2016), which are based on the thermodynamic and physical mechanisms by which BC is formed and oxidized. More specifically, the FOX

method requires the overall pressure ratio of each engine type as an input to estimate the BC $EI_m$, and we obtained this parameter from the ICAO Emission Data Bank (EDB). No smoke number measurements are required in the FOX and ImFOX methods. Since recently, the ICAO EDB provides non-volatile particle mass data; these are not used here because they are not available for older engines. The formulas and constants used to calculate the remaining parameters (GMD, GSD and $D_{fm}$) can be found in Teoh et al. (2020).

Lines 367 et seq.: This paragraph is an "aside" and perhaps did not receive as careful attention as the main conclusions. There are two statements in this paragraph that are not clearly stated.

The first sentence makes a point about fuel usage and aircraft types over Europe. The second sentence makes an additional point about fuel usage and aircraft types for a different case but does not explain the difference for this second set of statistics. Is it for a different geographic region (North America? The entire globe?)?

In the last sentence of this paragraph, the largest contrail contribution is noted. However, it is not clear if this is noting the largest contrail contribution for a single/individual airplane, or if it is the largest contribution to the total contrail impact of the fleet. The latter seems to not be the case, because of the prior statement about the twin-engine medium sized airliner (and presumably that was for 2020 also?), but the sentence is not clearly stated.

Reply: Apparently, the text needs some clarification. The whole paragraph refers to the same set of data, all for Europe.
We now write:
As an aside, it was found that 80 % (90 %) of fuel consumption over Europe comes from just 15 (23) aircraft types, whilst 80 % (90 %) of the contrail forcing came from 13 (19) types in 2019 and from 16 (24) types in 2020.   One particular aircraft type, a twin-engine medium-sized airliner, produced nearly 20 % of total fuel consumption and 16 % of contrail forcing, in the same data set. The largest contrail contribution in 2020 came from one type of twin-engine heavy aircraft, probably as a result of the larger fraction of cargo flights in 2020  (ICAO, 2021).

**Further changes:**

The traffic input and monthly mean output data are made available in public domain (Schumann, 2021a, b).

Additional references cited:

 recent related study: Gettelman et al. (2021)

recent paper discussing efficacy: Ponater et al. (2021)

related to soot emission calculation: Abrahamson et al. (2016)

and  paper discussing ice supersaturation: Lamquin et al. (2012)

Also a few minor text improvements were added. See comparison.

**References cited in this reply:**

Abrahamson, J. P., J. Zelina, M. G. Andac, and R. L. Vander Wal: Predictive model development for
aviation black carbon mass emissions from alternative and conventional fuels at ground and cruise, Env. Sci. Techn., 50, 12048-12055, doi: 10.1021/acs.est.6b03749 2016.

Gettelman, A., C.-C. Chen, and C. G. Bardeen: The climate impact of COVID19 induced contrail changes, Atmos. Chem. Phys. Discuss. [preprint], 17, https://acp.copernicus.org/preprints/acp-2021-210/, 2021.

ICAO: Effects of novel coronavirus (COVID-19) on civil aviation: Economic impact analysis, https://www.icao.int/sustainability/Documents/Covid-19/ICAO_coronavirus_Econ_Impact.pdf, Montreal, 125 pp., 2021.

Lamquin, N., C. J. Stubenrauch, K. Gierens, U. Burkhardt, and H. Smit: A global climatology of upper-tropospheric ice supersaturation occurrence inferred from the Atmospheric Infrared Sounder
calibrated by MOZAIC, Atmos. Chem. Phys., 12, 381-405, doi: 10.5194/acp-12-381-2012, 2012.

Ponater, M., M. Bickel, L. Bock, and U. Burkhardt: Towards determining the contrail cirrus efficacy, Aerospace, 8, 1-10, doi: aerospace8020042, 2021.

Sanz-Morère, I., S. D. Eastham, F. Allroggen, R. L. Speth, and S. R. H. Barrett: Impacts of multi-layer overlap on contrail radiative forcing, Atmos. Chem. Phys., 21, 1649-1681, doi: 10.5194/acp-21-1649-
2021, 2021.

Schumann, U.: Air traffic during COVID-19 over Europe [Data set]. Zenodo. http://doi.org/10.5281/zenodo.4661737,2021a.

Schumann, U.: Aviation contrail cirrus and radiative forcing over Europe for six months in 2020 during COVID-19 compared with 2019: Observations and model results (Version Version1) [Data set].
Zenodo. http://doi.org/10.5281/zenodo.4481680,2021b.

Stettler, M. E. J., A. M. Boies, A. Petzold, and S. R. H. Barrett: Global civil aviation black carbon emissions, Env. Sci. Techn., 47, 10397-10404, doi: 10.1021/es401356v, 2013.

Teoh, R., U. Schumann, A. Majumdar, and M. E. J. Stettler: Mitigating the climate forcing of aircraft contrails by small-scale diversions and technology adoption, Env. Sci. Techn., 54, 2941–2950, doi:
10.1021/acs.est.9b05608, 2020.

---

## Author Response (AR2)

Reply to Reviewer

Blue: reviewer's comments

Black: response

We thank the reviewer for his careful review.

The authors have made some important additions and modifications to the paper in response to the review, for which I thank them. Given that this is a second review, I have focused my comments on those areas which were changed (or not changed) in response to my prior assessment.

Firstly, I thank the authors for their detailed description of the changes in aircraft traffic composition during the 2020 pandemic (Tables 1 and 2 in the updated manuscript). This is valuable information and helps to support the later analysis. I also appreciate the changes made to the conclusions which discuss why the net RF was reduced by less (relatively) than either the shortwave or longwave RF, and the difficulty of drawing broader conclusions. The reference to a follow-up paper which discusses a comparison with satellite observations is also a valuable addition and helps to settle some of my prior issues.

Thank you.

I remain concerned that the paper is half focused on methodological advances in contrail modeling and half on the modeled changes in contrail cover (and RF) during 2020 compared to prior years, with no clear conceptual link. I accept that the authors have chosen not to split the paper. I also understand and agree with their assessment that the change in net RF due to an event (such as COVID) or intervention is somewhat unpredictable, due to the sensitive balance of longwave and shortwave RF. As such, I would recommend that the authors include a quantitative, comparative assessment of the specific contributions of each of their model advances to the nonlinearity in the response of the RF components with the COVID-induced change in flight distance (e.g. "after accounting for water exchange, the relative change in longwave RF between 2019 and 2020 is reduced by 10%. Contrail overlap modifies the change in longwave RF by..."). While multiple factors necessarily contribute to such changes, quantitative assessment of their contributions must at some level be possible. Such analysis would increase the scientific significance, the utility to the modeling community, and the coherence of the manuscript, by relating the model advances to the central result. This is currently only implied by data provided in single-month analyses such as in Table 8, which looks at the two years separately but do not quantify or discuss how the COVID-related reduction in RF is affected.

We thank the reviewer for his considerations. The reviewer's conclusion "that the paper is half focused on methodological advances in contrail modeling and half on the modeled changes in contrail cover (and RF) during 2020 compared to prior years", could be stated more positive: The paper addresses both, methodological advances in contrail modeling, and

modelling of changes due to the first 6 months of COVID-19. The conceptual link is stated in the Introduction, lines 64 ff and 78 ff: 1) to quantify air traffic activity, the related contrail cirrus and the radiative forcing for Europe in the months March to August 2019 and 2020; and 2) to describe the new traffic data set, its setup for 2019 and 2020, to quantify the changes in traffic, fuel consumption, soot emissions, contrail cover, RF and the related TOA irradiances, and test the sensitivity of the results to model parameters. - The reviewer asks for a more "quantitative, comparative assessment of the specific contributions of each of their model advances to the nonlinearity in the response of the RF components with the COVID-induced change in flight distance". On the other hand, the conclusions (lines 478 to 492 in the new revised version) explicitly quantify and discuss effects of changes in humidity, cloud overlap and soot. - Finally, we agree that more could be done, but this could be as well subject of future studies, possibly including longer periods, for which we are preparing right now. Therefore, we do not change the paper in this respect.

Minor comment: it would be helpful to include in the captions of Tables 5-8 that these refer only to results for July, as otherwise there is some confusion over the differing results when comparing (e.g.) Tables 3 and 8.

Response: We agree, and the time periods are now identified in all table captions.

Minor comment: there appears to be a formatting error Immediately after heading 5.2, with three lines given in boldface. See lines 432-434 in the document with tracked changes, or 422-424 of the clean manuscript.

Response: We agree, there was a formatting error. This got corrected.